**Air Quality Improvement in a Megacity: Implications from 2015 Beijing Parade**
**Blue Pollution-Control Actions**
Wen Xu[1,a,#], Wei Song[2,#], Yangyang Zhang[1,#], Xuejun Liu[1,*], Lin Zhang[3], Yuanhong Zhao[3],
Duanyang Liu[4], Aohan Tang[1], Daowei Yang[1], Dandan Wang[1], Zhang Wen[1], Yuepeng Pan[5], David
Fowler[6], Jeffrey L. Collett Jr.[7], Jan Willem Erisman[8], Keith Goulding[9], Yi Li[10], Fusuo Zhang[1]
[1]College of Resources and Environmental Sciences, Center for Resources, Environment and
Food Security, Key laboratory of Plant-Soil Interactions of MOE, China Agricultural University,
Beijing 100193, China
[2]Institute of Surface-Earth System Science, Tianjin University, Tianjin, 300072, China
[3]Laboratory for Climate and Ocean-Atmosphere Studies, Department of Atmospheric and
Oceanic Sciences, School of Physics, Peking University, Beijing 100871, China
[4]Jiangsu Meteorological Observatory, Nanjing 210008, China
[5]State Key Laboratory of Atmospheric Boundary Layer Physics and Atmospheric Chemistry
(LAPC), Institute of Atmospheric Physics, Chinese Academy of Sciences, Beijing 100029, China
[6]Centre for Ecology and Hydrology Edinburgh, Bush Estate, Penicuik, Midlothian EH26 0QB,
UK
[7]Department of Atmospheric Science, Colorado State University, Fort Collins, CO 80523, USA
[8]Louis Bolk Institute, Hoofdstraat 24, 3972 LA Driebergen, The Netherlands
[9]The Sustainable Soil and Grassland Systems Department, Rothamsted Research, West Common,
Harpenden, Hertfordshire, AL5 2JQ, UK
[10]Arizona Department of Environmental Quality, Phoenix, AZ, 85007, USA
[a]Current address: State Key Laboratory of Urban and Regional Ecology, Research Center for
Eco-Environmental Sciences, Chinese Academy of Sciences, Shuangqing Road 18, Haidian
District, Beijing 100085, China
[#] Equal contribution; [*] Corresponding author (Email: liu310@cau.edu.cn)
**Abstract:**
The implementation of strict emission control measures in Beijing and surrounding
regions during the 2015 China Victory Day Parade provided a valuable opportunity
to investigate related air quality improvements in a megacity. We measured $NH_3$,
$NO_2$ and $PM_{2.5}$ at multiple sites in and outside Beijing and summarized
concentrations of $PM_{2.5}$, $PM_{10}$, $NO_2$, $SO_2$ and CO in 291 cities across China from a
national urban air quality monitoring network between August and September 2015.
Consistently significant reductions of 12-35% for $NH_3$ and 33-59% for $NO_2$ in
different areas of Beijing city during the emission control period (referred to as the
Parade Blue period) were observed compared with measurements in the pre- and

post-Parade Blue periods without emission controls. Average $NH_3$ and $NO_2$ concentrations at sites near traffic were strongly correlated and showed positive and significant responses to traffic reduction measures, suggesting that traffic is an important source of both $NH_3$ and $NO_x$ in urban Beijing. Daily concentrations of $PM_{2.5}$ and secondary inorganic aerosol (sulfate, ammonium, and nitrate) at the urban and rural sites both decreased during the Parade Blue period. During (after) the emission control period, concentrations of $PM_{2.5}$, $PM_{10}$, $NO_2$, $SO_2$ and CO from the national city-monitoring network showed the largest decrease (increase) of 34-72% (50-214%) in Beijing, a smaller decrease (a moderate increase) of 1-32% (16-44%) in emission control regions outside Beijing, and an increase (decrease) of 6-16% (-2-7%) in non-emission control regions of China. Integrated analysis of modeling and monitoring results demonstrated that emission control measures made a major contribution to air quality improvement in Beijing compared with a minor contribution from favorable meteorological conditions during the Parade Blue period. These results show that controls of secondary aerosol precursors ($NH_3$, $SO_2$ and $NO_x$) locally and regionally are key to curbing air pollution in Beijing and probably in other mega cities worldwide.

**Introduction**

China's economy has made great advances over the last three decades. Its gross domestic production (GDP) ranked fifteenth in the world in 1978 but has risen to second place since 2010. During this period, environmental pollution has greatly increased, including soil, water and air pollution (Chan et al., 2008; Guo et al., 2010; Chen et al., 2014; Lu et al., 2015), which has become a major issue for the country. The Chinese government and people have grown particularly concerned about reducing air pollution since the large-scale haze pollution that occurred in China in January 2013. This episode affected an area of approximately 1.3 million $km^2$ and 800 million people (Huang et al., 2014). It led to serious human health problems and forced the Chinese government to address the problem of very large exposures of the

Chinese population to $PM_{2.5}$ (particulate matter ≤ 2.5 µm in aerodynamic diameter)
pollution. For example, compared with a similar winter period without haze
pollution (daily child patients < 600), more than 7000 daily child patients were
reported in Beijing Children's Hospital during the smog period in January 2013
(http://qnck.cyol.com/html/2014-01/01/nw.D110000qnck_20140101_1-28.htm).    In
response to this the 'Atmospheric Pollution Prevention and Control Action Plan' was
implemented by the Chinese government in September 2013, aiming to reduce $PM_{2.5}$
in Beijing by at least 25% from the 2012 level by 2017.
Many industrialized megacities have experienced severe air pollution, such as Los
Angeles during the 1940s-1970s (Haagen-Smit, 1952; Parrish et al., 2011), Mexico
city in the 1980s (Parrish et al., 2011), and London in the 1950s (Davis et al., 2002).
In these megacities, however, enormous progress in improving air quality has been
achieved with the implementation of various emission control strategies over recent
decades, despite rapid population growth and urbanization. According to Parrish et al.
(2011), first stage smog alerts in Los Angeles have decreased from some 200 per
year in the 1970s to about 10 per year now, and concentrations of air pollutants in
Mexico City have been reduced substantially over the past decades. Also, air quality
is now much better in London, with mean annual $PM_{10}$ (particulate matter ≤10 µm in
aerodynamic diameter) levels closer to 30 µg m$^{-3}$ than the 300 µg m$^{-3}$ fifty years ago
(and approx. 3000 µg m$^{-3}$ in December 1952) (Davis et al., 2002).
Beijing, the capital of China, is one of the largest megacities in the world with 22
million inhabitants and an area of 16800 square kilometers. The city is enclosed by
the Yanshan Mountains to the north and Taihang Mountains to the west. Its
fan-shaped topography permits efficient southerly transport of pollutants to Beijing,
which reduces air quality (Chen et al., 2015). A 70[th] anniversary victory parade was
held in Beijing on 3 September 2015 to commemorate the conclusion of the second
Sino-Japanese War and the end of World War II. The Chinese government imposed a
series of strict and urgent air pollutant emission-reduction measures to improve air
quality during what has been called the 'Parade Blue' period, from 20 August to 3
September 2015, in Beijing and surrounding regions (including Tianjin City, Inner
Mongolia Autonomous Region, Hebei, Shandong, Shanxi, and Henan Provinces) to
guarantee better air quality in the city. During this period, motor vehicles (except
taxis and buses) with even or odd registration numbers were banned on alternate
days, 1927 industrial enterprises had to limit production or were shut down, and
hundreds of construction sites in Beijing were closed, reducing air pollutant
emissions by 40% (http://gongyi.sohu.com/20150826/n419765215.shtml). For all
seven of the cities, provinces and autonomous regions, air pollutant emissions during
the Parade Blue period were decreased by 30% through a variety of reduction
measures (http://news.sohu.com/20150819/n419198051.shtml). No additional
pollution control measures were taken in other regions of China (outside Beijing and
surrounding regions) during this period.
Previous studies have attempted to quantify the role of short-term pollutant emission
control measures in air quality improvement in Beijing during the 2008 Olympics
(Wang et al., 2009, 2010; Shen et al., 2011) and the 2014 Asia-Pacific Economic
Cooperation (APEC) meeting (Chen et al., 2015). In addition, Tang et al. (2015)
reported that local emissions are the key factors determining the formation and
development of air pollution in the Beijing area. Ianniello et al. (2010) inferred that
traffic may be an important emission source of $NH_3$ in Beijing. However, the above
studies did not systematically answer the three following questions: what were (1)
the contribution of ammonia ($NH_3$) sources to urban $PM_{2.5}$ pollution; and (2) the
relative roles of pollution control measures and weather conditions in air quality
improvement? The present study attempts to examine these important topics by
taking advantage of the implementation of emission controls for the 70[th] anniversary
victory parade. We present results showing changes in concentrations of atmospheric
pollutants (i.e., $NH_3$, $NO_2$, $PM_{2.5}$ and associated inorganic water-soluble ions (WSIs))
before, during, and after the Parade Blue period, obtained from *in situ* measurements
at thirty-one sites in and outside Beijing. In addition, we compare the Chinese
Ministry of Environmental Protection officially released daily concentrations of
$PM_{2.5}$, $PM_{10}$, $NO_2$, $SO_2$ and CO at 291 cities in China during the same period. The
first results from the analysis of this extensive dataset reveal clear effects of the
Parade Blue emission reduction measures on air quality improvement and provide a
scientific basis for demonstrating the effectiveness of such control measures for air
pollution in mega cities.
**2 Materials and methods**
*2.1 Site selection and description*
Thirty-one air pollution monitoring sites have been established in and outside
Beijing municipality, with longitudes ranging from 115.02 ºE to 118.20 ºE and
latitudes from 36.84 ºN to 40. 34 ºN (**Fig. 1**). The 28 monitoring sites in Beijing
municipality are grouped into road and non-road sites to better distinguish the
impacts of control measures on sites near traffic. A brief description of all the sites is
given below. Detailed information, including specific sampling site, site type, and
potential emission sources for each site, is listed in **Table S1** in the Supplement.
*In Beijing*: Sixteen roadside monitoring sites are homogeneously distributed at the
edges of three major roads, including four sites each on the 3[rd] and 4[th] ring roads, and
eight sites on the 5[th] ring road. Additional road sites (sites 26 to 28) are in northwest
rural regions near the Yanshan mountains. Site 26 is located at the edge of the
Badaling highway, about 46 km northwest of the center of Beijing. Sites 27 and 28
are located, respectively, inside (100 m from the exit) and outside (30 m from the
entrance) the Badaling Highway Tunnel (1091.2 m long), which has two traffic
tunnels with one lane in each. The road sites were strongly and directly influenced
by vehicle emissions. Nine non-road sites were chosen over a wide area, extending
from an urban area (site 1) near the city center, through suburban areas (sites 6, 11,
12 and 13) between the 3[rd] and 5[th] ring roads, and ending in rural areas (sites 22 to
25) between the northwest 5[th] and 6[th] ring roads. These are likely to be polluted by
emissions from various sources, including dense housing, industry, cropland, small
villages, etc.
*Outside Beijing*: Site 29 is located in a rural area of Yucheng city, Shandong
province. Site 30 is located in Quzhou county, Hebei province, which is a typical
rural agricultural site with a recently constructed industrial district. Site 31 is a
regional background site located on Changdao island, Shandong province.
*2.2 Sampling procedure and sample analysis*
Atmospheric $NH_3$, $NO_2$ and $PM_{2.5}$ were measured from 3 August to 30 September
2015. The period can be divided into three phases: (1) 3-19 August (named
pre-Parade Blue period), (2) 20 August-3 September (Parade Blue period), and (3)
4-30 September (post-Parade Blue period). The sampling durations, measured
pollutants and number of samples for all the sites during each phase are summarized
in **Table S1** in the Supplement. The measurements of $NH_3$, $NO_2$ and $PM_{2.5}$ were not
concurrently made at most sites due to a shortage of manpower and samplers, but the
corresponding sampling sites together covered the major emission sources of
measured pollutants. Methods for sampling gases and $PM_{2.5}$ are briefly presented
below. For further details of the methodology the reader is referred to relevant
previous publications (Xu et al., 2014, 2015, 2016).
*Gaseous $NH_3$ and $NO_2$*: $NH_3$ samples were collected using ALPHA passive samplers
(Adapted Low-cost High Absorption, provided by the Centre for Ecology and
Hydrology, Edinburgh, UK) and $NO_2$ samples using Gradko diffusion tubes (Gradko
International Limited, UK). At each site, three ALPHA samplers and/or three $NO_2$
tubes were deployed under a PVC shelter (2 m above the ground) to protect the
samplers from rain and direct sunlight (Pictures for 4 selected road sites are shown in
**Fig. S1** of the Supplement). The samplers were exposed for 7 to 14 days during the
three study phases. $NH_3$ was extracted with high-purity water (18.2 MΩ) and analyzed
using a continuous-flow analyzer (Seal AA3, Germany). $NO_2$ samples, also extracted
with high-purity water, were analyzed using a colorimetric method by absorption at a
wavelength of 542 nm. More details of the passive samplers and their laboratory
preparation and analysis can be found in Xu et al. (2014, 2015).
*Airborne $PM_{2.5}$*: 24-h $PM_{2.5}$ samples were collected on 90 mm quartz fiber filters
(Whatman QM/A, Maidstone, UK) using medium-volume samplers (TH-150CIII,
Tianhong Co., Wuhan, China), at a flow rate of 100 L min$^{-1}$. The PM$_{2.5}$ mass was
determined using the standard gravimetric method, and one quarter of each PM$_{2.5}$
sample was ultrasonically extracted with 10 ml high-purity water for 30 min, with
the extract being filtered by a syringe filter (0.45 µm, Tengda Inc., China). The
water-soluble cations (NH$_4^+$, Na$^+$, Ca$^{2+}$, K$^+$, Mg$^{2+}$) and anions (NO$_3^-$, SO$_4^{2-}$, F$^-$, Cl$^-$)
in the extract were analyzed using Dionex-600 and Dionex-2100 Ion
Chromatographs (Dionex Inc., Sunnyvale, CA, USA), respectively (Zhang et al.,
2011; Tao et al., 2014).
*2.3 Quality assurance/ Quality control (QA/QC)*
All samples were prepared and measured in the Key Laboratory of Plant-Soil
Interactions, Chinese Ministry of Education, China Agricultural University, which
has a complete and strict quality control system. Three field (travel) blanks were
prepared for each batch of samples and analyzed together with those samples. All
reported concentrations of gases and PM$_{2.5}$ mass and ion concentrations are corrected
for the blanks. The detection limits were 0.01-0.02 mg L$^{-1}$ for the measured ions.
The measurement precisions were in the range of 5-10% for NH$_3$, NO$_2$, PM$_{2.5}$ mass
and water soluble ion concentrations. Quality assurance was routinely (once every
15-20 samples) checked using standard (designed specific concentrations of various
ions) samples during sample analysis.
*2.4 Other data collection*
The 24-h (daily) average concentrations of PM$_{2.5}$, PM$_{10}$, NO$_2$, SO$_2$ and CO measured
in 291 cities across China (including Beijing city, surrounding 63 cities in emission
control regions (hereafter termed to emission control regions (excluding Beijing)),
and 227 cities in other regions of China (hereafter referred to as non-emission
control regions) during the Pre-Parade Blue, Parade Blue and post-Parade periods
were downloaded from the Ministry of Environmental Protection (MEP) of China
(http://www.mep.gov.cn). These data for each city are summarized in **Tables S2-6** in
the Supplement. For Beijing city, each pollutant's daily individual Air Quality Index
(AQI) during the above three periods was calculated from the 24-h average
concentration. The highest individual AQI was selected and used as the daily AQI.
An AQI of 0-50, 51-100, 101-150, and 151-200 is classified as "excellent", "good",
"slightly polluted" and "moderately polluted", respectively. Details of the
calculations of AQI and the associated classification of air quality are given in the
Chinese Technical Regulations on AQI (MEPC, 2012).
Daily meteorological data in the above mentioned 291 cities for wind speed (WS),
temperature ($T$), and relative humidity (RH) during the Parade Blue period and
non-Parade Blue periods (the pre- and post-Parade Blue periods) were obtained from
Weather Underground (http://www.underground.com). The daily precipitation and
half-hourly wind speed and direction were measured in Beijing city. The
NCEP/NCAR global reanalysis meteorological data (including daily wind speed,
wind direction, sea surface pressure and precipitation) during the same periods were
provided by the NOAA/OAR/ESRL PSD, Boulder, Colorado, USA, from their
website (http://www.esrl.noaa.gov/psd). The daily mean atmospheric mixing layer
height (MLH) in Beijing during the period from 3 August to 30 September 2015 was
calculated using the method described in Holzworth (1964, 1967).

*2.5 Back trajectories and statistical analysis*
The 72-h (3-day) backward trajectories arriving at Beijing were calculated four times
a day (00:00, 06:00, 12:00, and 18:00 UTC) at 100 m height using the Hybrid Single
Particle Lagrangian Integrated Trajectory (HYSPLIT-4, NOAA) 4.9 model (Draxler
and Hess, 1997). Meteorological data with a resolution of $0.5° \times 0.5°$ were input
from the Global Data Assimilation System (GDAS) meteorological data archives of
the Air Resource Laboratory, National Oceanic and Atmospheric Administration
(NOAA). The trajectories were then grouped into four clusters during each period
using cluster analysis based on the total spatial variance (TSV) method (Draxler et
al., 2012). Values of $NH_3$, $NO_2$, $PM_{2.5}$ and ion concentrations per study phase at the
sampling sites are shown as the mean $\pm$ standard error (SE). Temporal differences
between study phases of concentrations of measured gases ($NH_3$ and $NO_2$) and the
MEP of reported pollutants (i.e. $PM_{2.5}$, $PM_{10}$, $NO_2$, $SO_2$ and CO) were investigated
using paired t-tests while those of measured $PM_{2.5}$ mass and associated ionic
components were investigated using a non-parametric Mann-Whitney U test. All
statistical analyses were performed using SPSS11.5 (SPSS Inc., Chicago, IL, USA).
Statistically significant differences were set at $p < 0.05$ unless otherwise stated.

**3. Results**

*3.1 Concentrations of gaseous $NH_3$ and $NO_2$*
Ambient $NH_3$ concentrations varied greatly during the pre-Parade Blue, Parade Blue
and post-Parade Blue periods, with values of 8.2-31.7, 7.8-50.7 and 7.4-40.2 $\mu g\ m^{-3}$,
respectively (**Fig. 2A a**). The average $NH_3$ concentrations during the three periods
for the sites inside the $6^{th}$ ring road (including road sites (RS) on the $3^{rd}$, $4^{th}$ and $5^{th}$
ring roads and non-road sites (NRS)), outside the $6^{th}$ ring road but in Beijing and
outside Beijing, are shown in **Fig. 2A b and c**. The mean $NH_3$ concentration inside
the $6^{th}$ ring road was significantly smaller (by 13%) during the Parade Blue period
compared with the mean during the post-Parade Blue period ($20.2 \pm 1.2\ \mu g\ m^{-3}$
versus $23.3 \pm 1.8\ \mu g\ m^{-3}$); further, on all three ring roads reductions (23 to 35%) of
the mean during the Parade Blue period were statistically significant while at the
non-road sites a small non-significant increase (15%) in the mean was observed (**Fig.**
**2A c**). The mean $NH_3$ concentration outside the $6^{th}$ ring road was 12% smaller in the
Parade Blue period than in the post-Parade Blue period ($21.4 \pm 6.0\ \mu g\ m^{-3}$ versus
$24.3 \pm 9.3\ \mu g\ m^{-3}$), whereas outside Beijing, non-significant increases (on average
80%) in the mean occurred during the Parade Blue period ($26.7 \pm 12.6\ \mu g\ m^{-3}$)
compared with those during the pre- and post-Parade Blue periods ($19.9 \pm 6.2$ and
$11.8 \pm 2.3\ \mu g\ m^{-3}$, respectively).
Ambient $NO_2$ concentrations ranged from 21.5 to 227.7, 14.1 to 258.8, and 15.7 to
751.8 $\mu g\ m^{-3}$ during the pre-Parade Blue, Parade Blue and post-Parade Blue periods,
respectively (**Fig. 2B a**). The mean $NO_2$ concentrations at the sites inside the $6^{th}$ ring
road (including road sites on the $5^{th}$ ring road and NRS), outside the $6^{th}$ ring road and
outside Beijing during the three periods are shown in **Fig. 2B b and c**. Inside the $6^{th}$
ring road, the mean concentration during the Parade Blue period (78.7 µg m$^{-3}$) was
42% and 35% lower ($p<0.01$) than the means during the pre- and post-Parade Blue
periods (135.7 ± 21.8 and 121.0 ± 16.5 µg m$^{-3}$, respectively). For the $5^{th}$ ring RS and
NRS, most reductions (33~42%) in the mean during the Parade Blue period were
also highly significant ($p<0.01$). Inside the $6^{th}$ ring road, a large non-significant
reduction (59%) in the mean concentration occurred during the Parade Blue period
compared with the post-Parade Blue period (183.5 ± 49.1 versus 443.4 ± 173.3 µg
m$^{-3}$). Outside Beijing, the change in the mean during the Parade Blue period (23.7 ±
3.6 µg m$^{-3}$) was small and non-significant when compared with the means during the
pre- and post-Parade periods (27.5 ± 4.5 and 18.5 ± 1.7 µg m$^{-3}$, respectively).

*3.2 Concentrations of PM$_{2.5}$ and its chemical components*
A statistical analysis of concentrations of PM$_{2.5}$ mass and associated inorganic WSIs
at sites 22, 29 and 30 in the three periods is presented in **Table 1**. Daily PM$_{2.5}$
concentrations ranged from 4.2 to 123.6, 15.4 to 116.0, and 12.4 to 170.7 µg m$^{-3}$ at
sites 22, 29 and 30, respectively. At sites 22 and 29, mean PM$_{2.5}$ concentrations
during the Parade Blue period decreased significantly (by 49% and 40%,
respectively) compared with the means during the pre-Parade Blue period, and
increased again during the post-Parade Blue period (57% and 3%, respectively)
compared with the means during the Parade Blue period. At site 30, a 24% reduction
in mean PM$_{2.5}$ concentrations occurred during the Parade Blue period compared with
the pre-Parade Blue period and a 103% increase during the post-Parade Blue period.
Secondary inorganic aerosols (SIA) (sum of NH$_4^+$, NO$_3^-$ and SO$_4^{2-}$) were major
components of PM$_{2.5}$, with average contributions of 24%, 41% and 32% to the daily
average PM$_{2.5}$ mass at sites 22, 29 and 30, respectively. As with PM$_{2.5}$
concentrations, concentrations of all the WSIs (except for Cl$^-$) at site 22 decreased
significantly during the Parade Blue period compared with the pre- and/or
post-Parade Blue periods. Analogous behavior also occurred at sites 29 and 30 for
concentrations of $NO_3^-$, $NH_4^+$ and $SO_4^{2-}$ but not for those of most of other ions (e.g.
$Ca^{2+}$, $K^+$, $F^-$, $Na^+$).

*3.3 Daily mean pollutant concentrations from MEP*
Daily mean concentrations of the five major pollutants ($PM_{2.5}$, $PM_{10}$, $NO_2$, $SO_2$ and
CO) at 291 cities in China, divided into three groups of Beijing, cities in emission
control regions (excluding Beijing) and cities in non-emission control regions, are
summarized in **Fig. 3**. Average concentrations of $PM_{2.5}$, $PM_{10}$, $NO_2$, $SO_2$ and CO
during the Parade Blue period were highly significantly ($p<0.01$) decreased in
Beijing, with reductions of 72%, 67%, 39%, 34% and 39%, respectively, compared
with the pre-Parade Blue period. $PM_{2.5}$ concentrations in Beijing, for example,
remained below 20 µg m$^{-3}$ for 14 consecutive days in the Parade Blue period (for
comparison: the WHO and China's (first-grade) thresholds for daily $PM_{2.5}$
concentrations are 25 and 35 µg m$^{-3}$, respectively). The daily $PM_{2.5}$ concentrations in
Beijing in the pre-Parade Blue period averaged 59 µg m$^{-3}$. Concentrations of $PM_{2.5}$,
$PM_{10}$ and $SO_2$ in the Parade Blue period were also significantly ($p<0.05$) decreased
in cities in emission control regions (excluding Beijing), with reductions of 32%,
29% and 7%, respectively, while concentrations of $NO_2$ and CO did not show
statistically significant changes ($p>0.05$). After the Parade Blue period,
concentrations of the five major pollutants in Beijing and surrounding regions
rebounded quickly, with significant increases of 50-214%, and 16-44%, respectively.
In cities in other regions, by contrast, where no additional emission reduction
measures were taken, concentrations of $PM_{2.5}$, $PM_{10}$, $NO_2$, $SO_2$ and CO remained
stable or were significantly ($p<0.05$) higher during the Parade Blue period compared
with the pre- and post-Parade Blue periods.

**4. Discussion**
*4.1 Effect of emission controls on air quality*
The statistical analyses (**Fig. 3**) show that, by taking regional emission controls
during the Parade Blue period, daily concentrations of the five reported pollutants
($PM_{2.5}$, $PM_{10}$, $NO_2$, $SO_2$ and CO) in Beijing city and surrounding other cities in the
six provinces were decreased by various but statistically significant amounts, in
sharp contrast to increases in cities in other parts of China where no additional
emission controls were imposed. This shows the effectiveness of the pollution
controls and suggests that air quality improvement was directly related to the
reduction intensities of pollutant emissions (e.g., air pollution control effects ranked
by Beijing (largest reduction) > emission control regions surrounding Beijing
(moderate reduction) > other regions (no reduction) in China). Another way of
quantifying the effect of the additional control measures for Beijing uses the Air
Quality Index (MEPC, 2012). On the basis of the calculated air quality index (AQI,
**Fig. 4**), defined "good" and polluted days (i.e. "slightly polluted" and "moderately
polluted") altogether accounted for 89% during the pre-Parade Blue period, and 70%
during the post-Parade Blue period. The primary pollutant was $PM_{2.5}$ for 82% and
63% of these days during the Pre- and post-Parade Blue periods, respectively. In
contrast, almost all of the days during the Parade Blue period were defined as
"excellent". Thus improved air quality-as represented by the AQI during the Parade
Blue period was mainly attributed to the additional control of $PM_{2.5}$ precursors.
Results from the MEP of source apportionment of $PM_{2.5}$ for Beijing
(http://www.bj.xinhuanet.com/bjyw/2014-04/17/c_1110289403.htm) showed that
64-72% of atmospheric $PM_{2.5}$ during 2012-2013 was generated by emissions from
local sources, of which the biggest contributor was vehicle exhaust (31.1%),
followed by coal combustion (22.4%), industrial production (18.1%), soil dust
(14.3%) and others (14.1%). The contribution from vehicles had increased by 1.7
percentage points compared to 2010-2011. To examine the contribution of vehicles,
power plants, and industries to $PM_{2.5}$ concentrations, $PM_{2.5}$ concentrations from
these were compared with those of other primary pollutants such as $NO_x$ ($NO+NO_2$),
CO and $SO_2$ (Zhao et al., 2012). As shown in **Fig. S2a-d** in the Supplement, the
linear correlations of $PM_{2.5}$ with each pollutant gas ($CO$, $NO_2$ and $SO_2$) and their sum
were positive and highly significant ($R=0.40$-$0.88$, $p<0.05$) during the study period,
except for the relationship between $PM_{2.5}$ and $NO_2$ during the pre-Parade Blue
period and that of $PM_{2.5}$ versus $SO_2$ during the Parade Blue period, both of which
were positive but not significant ($p>0.05$). This finding is consistent with the source
apportionment results that suggest traffic, power plants and industry are significant
sources of $PM_{2.5}$ in Beijing. Given the importance of local vehicle emissions vs.
more distant power plant and industrial emissions for Beijing's air quality, the ratio
of $CO/SO_2$ can be used as an indicator of the contribution of local emissions to air
pollution, with higher ratios indicating higher local contributions (Tang et al., 2015).
Ratios of $CO/SO_2$ decreased (on average by 18%) from the pre-Parade Blue to
Parade Blue period, and then increased abruptly on September 4[th] in the Post-Parade
Blue period (**Fig. 4**), further suggesting the decreased amount of pollutants from
local contributions. Beijing has relatively little industry but numerous automobiles,
and the emissions of $SO_2$ are small while those of $CO$ and $NO_x$ are much larger
(Zhao et al., 2012). Thus, traffic emission is likely to be a determining factor
influencing urban $CO$ and $NO_x$ levels. This, in combination with a strong positive
and highly significant correlation of $PM_{2.5}$ versus $CO+NO_2$ during the study period
(**Fig. S2e, Supplement**), and the weak correlation of $PM_{2.5}$ versus $SO_2$ noted above
(**Fig. S2c, Supplement**), shows that traffic emission controls should be a priority in
mitigating $PM_{2.5}$ pollution in the future.
Concentrations of $PM_{2.5}$ levels in Beijing are not only driven by primary emissions
but are also affected by meteorology and atmospheric chemistry operating on the
primary pollutants, leading to secondary pollutant formation (Zhang et al., 2015). To
quantify the likely contribution of secondary pollutant formation of $PM_{2.5}$ as a
contributor to the observed changes between the Parade Blue period and pre- and
post-measurements, $CO$ provides an excellent tracer for primary combustion sources
(de Gouw et al., 2009). Daily ratios of $PM_{2.5}/CO$ during the Parade Blue period
decreased significantly on average by 50% and 40% relative to the pre- and
post-Parade Blue periods, respectively (**Fig. 4**), which suggests that the significant
reduction of $PM_{2.5}$ concentrations during the Parade Blue period was not only due to
less anthropogenic primary emissions but also due to reduced secondary pollutant
formation. This is further supported by our measured results at urban site 22, where
average SIA concentrations comprised 20-29% of average $PM_{2.5}$ mass over the three
periods, and decreased significantly during the Parade Blue period compared with
those during the pre- and post-Parade Blue periods (**Table 1**). Significant reduction
of concentrations of precursor gases (e.g. $NO_2$, $SO_2$ and $NH_3$) at the city scale is
likely to be the major reason for such reduced secondary pollutant formation. In
addition, lower concentrations of sulfate and nitrate during the Parade Blue period
might also be caused by lower oxidation rates of $SO_2$ and $NO_x$. The sulfur oxidation
ratio    $(SOR=nSO_4^{2-}/(nSO_4^{2-}+nSO_2))$    and    the    nitrogen    oxidation    ratio
$(NOR=nNO_3^-/(nNO_3^-+nNO_2))$ ($n$ refers to the molar concentration) are indicators of
secondary pollutant transformation in the atmosphere. Higher values of SOR and
NOR imply more complete oxidation of gaseous species to sulfate- and
nitrate-containing secondary particles (Sun et al., 2006). To understand the potential
change in the degree of oxidation of sulfur and nitrogen, we used daily
concentrations of $SO_4^{2-}$ and $NO_3^-$ measured at urban site 22 (located at west campus
of China Agricultural University) and the MEP-reported concentrations of $SO_2$ and
$NO_2$ at the Wanliu monitoring station to calculate the SOR and NOR values. This is
because these two sites, only 7 km apart (**Fig. S3, Supplement**), experience similar
pollution climates. The average values of SOR and NOR were 0.64 and 0.04 during
the pre-Parade Blue period, 0.47 and 0.03 during the Parade Blue period, and 0.48
and 0.07 during the Post-Parade Blue period, respectively. (**Fig. S4, Supplement**).
Compared with the pre- and post-Parade Blue periods, slightly reduced values of
SOR and NOR during the Parade Blue period suggests a possible minor role for
changes in the extent of photochemical oxidation in secondary transformation.
Ammonia is the primary alkaline trace gas in the atmosphere. In ammonia-rich
environments, $NH_4HSO_4$ and $(NH_4)_2SO_4$ are sequentially formed, and the surplus
$NH_3$ that does not react with $H_2SO_4$ can form $NH_4NO_3$ (Wang et al., 2005). In both
the pre-Parade Blue and Parade Blue periods, $NH_4^+$ was strongly correlated with
$SO_4^{2-}$ (**Fig. S5 a** and **c, Supplement**) and $[SO_4^{2-}+NO_3^-]$ (**Fig. S5 b** and **d,**
**Supplement**), and the regression slopes were both 0.87 during the pre-Parade Blue
period, 0.97 and 0.91 during the Parade Blue period, and 1.13 and 0.79 during the
post-Parade Blue period, respectively. These results indicate almost complete
neutralization of acidic species ($HNO_3$ and $H_2SO_4$) by $NH_3$ in $PM_{2.5}$ during these
three periods especially in the Parade Blue period. In this way, SIA concentrations
from these sources could not be further reduced during the Parade Blue period unless
$NH_3$ emissions were reduced more than those of $SO_2$ and $NO_x$.
*4.2 Impact of traffic $NH_3$ emission on urban $NH_3$ concentration*
The sources of $NH_3$ are often dominated by agriculture, but it may also be produced
by motor vehicles due to the over-reduction of NO in catalytic converters (Kean et
al., 2000). The contribution of traffic to the total $NH_3$ emissions is estimated at
approximately 2% in Europe (EEA, 2011) and 5% in the US (Kean et al., 2009). In
China, $NH_3$ emissions from traffic rose from 0.005 Tg (contributing approximately
0.08% to total $NH_3$ emissions) in 1980 to 0.5 Tg (contributing approximately 5% to
total emissions) in 2012 (Kang et al., 2016). Recent studies have discussed the origin
of atmospheric $NH_3$ in Beijing city based on the $\delta^{15}N$ technique (Chang et al., 2016;
Pan et al., 2016). For example, Chang et al. (2016) identified that non-agricultural
sources, merged with waste and traffic $NH_3$ emissions, collectively accounted for
approximately 50% of ambient $NH_3$ in urban Beijing before and after APEC summit,
of which more than 20% was sourced from traffic emissions. Traffic is therefore
likely to make a very significant contribution to $NH_3$ concentrations in urban areas of
Beijing, and a strong correlation of $NH_3$ with traffic-related pollutants was observed
($NO_x$ and CO) at the urban sites (Ianniello et al., 2010; Meng et al., 2011). However,
this relationship has a large uncertainty because the concentrations of pollutants used
to establish the relationship were measured at 'background' urban sites some
distance from major roads, and other urban sources such as decaying organic matter
may contribute. In the present study, strong and significant correlations were
observed between $NH_3$ and $NO_2$ concentrations measured on the $5^{th}$ ring road during
all three periods (**Fig. 5**). In addition, compared with the averages for the three ring
roads during the pre- and/or post-Parade Blue periods, the average $NH_3$
concentrations during the Parade Blue period decreased significantly owing to traffic
reduction measures (**Fig. 2A c**). These results provide strong evidence that traffic is
an important source of $NH_3$ in Beijing. In addition to period-to-period temporal
changes, the mean $NH_3$ concentration at all road sites was 1.3 and 1.9 times that at
all non-road sites during the Parade Blue period and post-Parade Blue period,
respectively (**Fig. 2A**). Moreover, during the post-Parade Blue period the measured
$NH_3$ concentrations on the three ring roads ($28.3 \pm 6.4$ μg m$^{-3}$) were twice those at
the rural sites 29 and 30 ($14.0 \pm 1.6$ μg m$^{-3}$) affected by intense agricultural $NH_3$
emissions. These results, along with the fact that urban Beijing has a higher relative
on-road vehicle density and almost no agricultural activity, suggest that $NH_3$
emission and transport from local traffic were the main contributors to high urban
$NH_3$ concentrations. Based on a mileage-based $NH_3$ emission factors of $28 \pm 5$
(assumed as the lower limit, Chang et al., 2016) and $230 \pm 14.1$ mg km$^{-1}$ (assumed as
the upper limit, Liu et al., 2014) for light-duty gasoline vehicles, a population of 5.61
million vehicles (average mileage 21849 km vehicle$^{-1}$ yr$^{-1}$) in Beijing would produce
approximately 3.4-28 kt $NH_3$ in 2015, which likely declined by up to 4.7-38 t $NH_3$
day$^{-1}$ during the Parade Blue period, given that the traffic load decreased by half with
the implementation of the odd-and-even car ban policy. For accurately determining
$NH_3$ emissions, however, further study on $NH_3$ emission factors for vehicles and
other sources is warranted.

*4.3 Impact of meteorological conditions and long-range air transport*
Meteorological conditions strongly regulate near-surface air pollutant concentrations
(Liu et al., 2015), contributing the largest uncertainties to the evaluation of the
effects of emission controls on pollutant reduction. Here we first compared the
meteorological data obtained during the Parade Blue period with those from the pre-
and/or post-Parade Blue periods in Beijing and other cities over North China. In
Beijing, neither wind speed (WS) nor relative humidity (RH) differed significantly
between non-Parade Blue (the pre- and post-Parade Blue) and the Parade Blue
periods, while temperature ($T$) showed a significant but small decrease with time
(**Fig. 6**). Similarly, there were small and non-significant changes in $T$, WS and RH
between the pre-Parade Blue and Parade Blue periods for emission control regions
(excluding Beijing) and for non-emission control regions in China. These results
suggest that the period-to-period changes in $T$, WS and RH may have only a minor
impact on $PM_{2.5}$, $PM_{10}$, $NO_2$, $SO_2$ and CO concentrations in the emission control
regions (**Fig. 3**). In contrast, a higher temperature during the Parade Blue period,
compared to the post-Parade Blue period, can in part explain the corresponding
higher $NH_3$ concentrations measured at NRS, due to increased $NH_3$ emissions from
biological sources such as humans, sewage systems and organic waste in garbage
containers (Reche et al., 2012).
Surface weather maps of China (**Fig. S6, Supplement**) and North China (**Fig. 7**)
showed an apparent change of wind field over Beijing and its surrounding regions
during the Parade Blue period compared with the other two periods. As shown in **Fig.**
**7**, Beijing was located at the rear of a high pressure system within the
southeast/south flow or in a high-pressure area when the wind was weak ($< 3$ m s$^{-1}$),
and at the base of the Siberian high pressure system when influenced by a weak cold
front and easterly wind ($> 4$ m s$^{-1}$) in the non-Parade (pre- or post-Parade) Blue and
Parade Blue periods, respectively. The former weather condition (non-Parade Blue
periods) was conducive to pollutant convergence and the latter (Parade Blue period)
was conducive to pollutant dispersion. A further analysis of wind rose plots (**Fig. 8a**)
showed that northerly winds, with similar wind speeds, dominated all three periods.
Northerly/northwesterly winds in Beijing bring relatively clean air due to a lack of
heavy industry in the areas north/northwest of Beijing. Winds during the pre- and
post-Parade Blue periods were occasionally from the south, southeast and east of

Beijing, where the regions (e.g. Hebei, Henan and Shandong provinces) are characterized by substantially higher anthropogenic emissions of air pollutants such as $NH_3$, $NO_x$, $SO_2$ and aerosols (Zhang et al., 2009, 2010; Gu et al., 2012). Also as mentioned earlier, the topography of the mountains to the West and North of Beijing effectively traps the polluted air over Beijing during southerly airflow, suggesting that the southerly wind during non-Parade Blue periods may enhance air pollution in Beijing. Wet scavenging from precipitation, although often important in summer (Yoo et al., 2014), probably played a minor role in changing the concentrations of pollutants given the low and comparable precipitation over Beijing and surrounding areas during the study periods (**Fig. 8**). For example, the total precipitation in Beijing was comparable between the pre-Parade Blue and Parade Blue periods (38.9 versus 34.4 mm) (**Fig. 8b**). In addition, we compared daily mean mixing layer height (MLH) in Beijing during the study period (**Fig. 9a**). The daily mean MLH in Beijing was approx. 37% higher during the Parade Blue period (1777 m) than the pre-Parade (1301 m) and post-Parade (1296 m) Blue periods (**Fig. 9b**, $p = 0.08$). Since the MLH during Parade Blue was higher than that during non-Parade Blue periods, the horizontal and vertical diffusion conditions during the Parade Blue period were better than the other two periods.

Changes in meteorological conditions often lead to changes in regional pollution transport and ventilation. It has been shown that regional transport from neighboring Tianjin, Hebei, Shanxi, and Shandong Provinces can have a significant impact on Beijing's air quality (Meng et al., 2011; Zhang et al., 2015). Model calculations by Zhang et al. (2015) suggested that about half of Beijing's $PM_{2.5}$ pollution originates from sources outside of the city. Trajectory analysis in previous studies revealed that the air mass from south and southeast regions of Beijing led to high concentrations of $NH_3$, $NO_x$, $PM_{2.5}$ and secondary inorganic ions during summertime (Ianniello et al., 2010; Wang et al., 2010; Sun et al., 2015). The 72-hour back trajectories during the three measurement periods, shown in **Fig. 10**, were classified into 4 sectors according to air mass pathways: the west pathway over southern Mongolia, western

Inner Mongolia, and SinKiang, the north pathway over Inner Mongolia, Heilongjiang and north Hebei Provinces, the east pathway mainly over northeast Hebei province and Tianjin municipality, and the south sector over the south Hebei and Shandong provinces. The results indicated that transport of regional pollution from the south sector occurred during the pre-Parade Blue period (38%) and the post-Parade Blue period (18% for $PM_{2.5}$ sampling days and 29% for $NH_3$ sampling days) but there was no transport of regional pollution during the Parade Blue period. As the south of Hebei province contains heavily polluting industry and intensive agriculture (Zhang et al., 2009; Sun et al., 2015), the absence of transport of air masses from the south sector is likely at least partly responsible for lower concentrations of the five reported pollutants ($PM_{2.5}$, $PM_{10}$, $NO_2$, $SO_2$ and CO) during the Parade Blue period. As for $NH_3$, however, average concentration at NRS were slighter higher in the Parade Blue period than in the post-Parade Blue period (**Fig. 2A c**), indicating that surface levels of $NH_3$ were less influenced by southern air masses. Much of the airflow travelled over Tianjin municipality during the Parade Blue period (32%) compared to that during the post-Parade Blue period (19%) (**Fig. 10 b**, **d**), which probably caused the high surface $NH_3$ concentrations in Beijing. This is because Tianjin, as one of the mega-cities in China, has high $NH_3$ emissions from livestock and fertilizer application (Zhang et al., 2010).

To further diagnose the impacts of meteorology on the surface air quality, we conducted a simulation using the nested GEOS-Chem atmospheric chemistry model driven by the GEOS-FP assimilated meteorological fields at $1/4° \times 5/16°$ horizontal resolution covering East Asia (70°E-140°E, 15°N-55°N) (Zhang et al., 2015; 2016). Details of the model emissions and mechanisms have been described in Zhang et al. (2016), focusing on $PM_{2.5}$ concentrations in Beijing during the Asia-Pacific Economic Cooperation Summit (APEC; November 5-11) period. We used anthropogenic emissions from the Multi-Resolution Emission Inventory of China for the year 2010 (MEIC, 2015), except for $NH_3$ emissions that were taken from the Regional Emission in Asia (REAS-v2) inventory (Kurokawa et al., 2013) with an

improved seasonality derived by Zhao et al. (2015).
We conducted a standard simulation with fixed anthropogenic emissions for the
period of 1 August – 12 September 2015. By fixing anthropogenic emissions in the
simulation, the model provides a quantitative estimate of the meteorological impacts
alone before and during the Parade Blue period. For the pre-Parade period (1-19
August), the model-simulated mean $PM_{2.5}$ concentration is 62 µg m$^{-3}$ in Beijing,
comparable to the measured values (59 µg m$^{-3}$), but simulated $NH_3$ concentrations
are too low (3 µg m$^{-3}$ vs. 8.2-31.7 µg m$^{-3}$), probably due to missing urban $NH_3$
sources and the coarse model resolution ($1/4° \times 5/16°$). Here we focus on the model
simulated relative changes in pollutant concentrations before and during the Parade
Blue period. Model results showed that, without emission controls, the air pollutant
concentrations in Beijing in the Parade Blue period relative to the pre-Parade period
would be 29% lower for $PM_{2.5}$, 7% lower for $NH_3$, 17% lower for $SO_2$, 8% lower for
CO and relatively no change for $NO_2$ (**Fig. 11a**) as a result of the different
meteorological conditions as discussed above. Thus, compared with meteorological
condition changes (MCC), air pollution control measures (PCM) made a greater
contribution to air quality improvement (especially for $PM_{2.5}$, $NO_x$, and CO) in
Beijing during the Parade Blue period (**Fig. 11b**).
We also conducted two sensitivity simulations ((1) with anthropogenic emissions of
$NH_3$ reduced by 40% over Beijing and by 30% over Hebei and Tianjin; and (2) with
all anthropogenic emissions including $NH_3$, $SO_2$, $NO_x$, CO, and primary aerosol
reduced by 40% over Beijing and by 30% over Hebei and Tianjin) for the Parade
Blue period (20 August-3 September) to examine the responses of $PM_{2.5}$
concentrations to emission reductions. We find that the $NH_3$ emission reduction (by
40% over Beijing and by 30% over Hebei and Tianjin) could decrease the mean
$PM_{2.5}$ concentration in Beijing by 12% for the period, compared with 31% simulated
$PM_{2.5}$ reduction if all anthropogenic emissions were reduced by the same amount.
This supports our findings on the effectiveness of emission controls during the
Parade Blue period as indicated in the measurements, and the high sensitivity of
$PM_{2.5}$ concentration in Beijing to $NH_3$ sources.

*4.4 Implications for regional air pollution control*
Besides Tianjin, Beijing city is surrounded by four provinces, Hebei, Shandong,
Henan and Shanxi, which all have major power plants and manufacturing industry.
In the INTEX-B emission inventory, the total emissions from these four provinces
accounted for 28.7%, 27.9%, 28.3%, and 25.0% of national emissions of $PM_{2.5}$,
$PM_{10}$, $SO_2$, and $NO_x$, respectively (Zhang et al., 2009). The 'Parade Blue' experience
demonstrates that, by taking appropriate but strict coordinated regional and local
emission controls, air quality in megacities can be significantly and quickly
improved.
China is not the first country to use temporal emission control strategies. In 1996, the
city of Atlanta, for example, adopted a series of actions to reduce traffic volume and
significantly improved air quality during the Atlanta Olympic Games (Tian and
Brimblecombe, 2008; Peel et al., 2010). We also should note that most of these
emission control strategies have not been maintained after the Olympic Games. In
the long term, temporary emission control strategies will not improve regional air
quality conditions and we should seek better ways towards sustainable development.
Integrated emission reduction measures are therefore necessary, but meteorological
conditions also need to be considered for a sustainable solution, as in Urumqi,
northwest China (Song et al., 2015). We therefore recommend further efforts to build
on the Parade Blue experience of successful air quality improvement in Beijing and
the surrounding region to improve air pollution control policies throughout China
and in other rapidly developing countries.
Chinese national $SO_2$ emissions have been successfully reduced by 14% from the
2005 level due to an $SO_2$ control policy (Wang et al., 2014), and nationwide controls
on $NO_x$ emissions have been implemented along with the controls on $SO_2$ and
primary particles during 2011-2015. However, there is as yet no regulation or policy
that targets national $NH_3$ emissions. Future emission control policies to mitigate PM
and SIA pollution in China should, in addition to focusing on primary particles, $NO_x$
and $SO_2$, also address $NH_3$ emission reduction from both agricultural and
non-agricultural sectors (e.g. traffic sources) in particular when $NH_3$ becomes key to
$PM_{2.5}$ formation (Liu et al., 2013; Wu et al ., 2016; Xu et al., 2016).

**Conclusions**
We have presented atmospheric concentrations of $NH_3$, $NO_2$, $PM_{2.5}$ and associated
inorganic water-soluble ions before, during, and after the Parade Blue period
measured at thirty-one *in situ* sites in and outside Beijing, and daily concentrations
of $PM_{2.5}$, $PM_{10}$, $NO_2$, $SO_2$ and CO in 291 cities in China during the pre-Parade Blue
and Parade Blue periods released by the Ministry of Environmental Protection (MEP)
of China. Our unique study examines temporal variations at local and regional scales
across China and the relative role of the emission controls and meteorological
conditions, as well as the contribution of traffic, to $NH_3$ levels in Beijing based on
the first direct measurements of $NH_3$ and $NO_2$ concentrations at road sites. The
following major findings and conclusions were reached:
The concentrations of $NH_3$ and $NO_2$ during the Parade Blue period at the road sites
in different areas of Beijing decreased significantly by 12-35% and 34-59%
respectively relative to the pre-and post-Parade Blue measurements, while those at
the non-road sites showed an increase of 15% for $NH_3$ and reductions of 33% and
42% for $NO_2$. Positive and significant correlations were observed between $NH_3$ and
$NO_2$ concentrations measured at road sites. Taken together, these findings indicate
that on-road traffic is an important source of $NH_3$ in the urban Beijing. Daily
concentrations of $PM_{2.5}$ and secondary inorganic aerosols (sulfate, ammonium, and
nitrate) at the urban and rural sites both decreased during the Parade Blue period,
which was closely related to controls of secondary aerosol precursors ($NH_3$, $SO_2$ and
$NO_x$) and/or reduced secondary pollutant formation.
During the Parade Blue period, daily concentrations of air pollutants ($PM_{2.5}$, $PM_{10}$,
$NO_2$, $SO_2$ and CO) in 291 cities obtained from the national air quality monitoring

network showed large and significant reductions of 34-72% in Beijing, small reductions of 1-32% in emission control regions (excluding Beijing), and slight increases (6-16%) in non-emission control regions that in some cases were significant, which reflects the positive effects of emission controls on air quality and suggests that the extent of air quality improvement was directly associated with the reduction intensities of pollutant emissions.

A detailed characterization of meteorological parameters and regional transport demonstrated that the good air quality in Beijing during the Parade Blue period was the combined result of emission controls, meteorological effects and the absence of transport of air masses from the south of Beijing. Thus, the net effectiveness of emission controls was investigated further by excluding the effects of meteorology in model simulations, which showed that emission controls can contribute reductions of pollutant concentrations of nearly 60% for $PM_{2.5}$, 109% for $NO_2$, 80% for CO, 53% for $NH_3$ and 50% for $SO_2$. This result showed that emission controls played an dominant role in air quality improvement in Beijing during the Parade Blue period.

**Acknowledgments**

We thank Lu Li, Hao Tianxiang, Wang Sen and Wang Wei for their assistance during the field measurements. This work was financially supported by the 973 project (2014BC954200) and the National Natural Science Foundation of China (41425007, 31421092).

**Author Contributions**

X.L. and F.Z. designed the research. X.L., W.X., W.S., Y.Z., D.Y., D.W. Z.W. and A.T. conducted the research (collected the data and performed the measurements). W.X., W.S., Z.L. and X.L. wrote the manuscript. All authors were involved in the discussion and interpretation of the data as well as the revision on the manuscript.

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

**Figure captions**

**Fig. 1**. Maps showing the thirty-one monitoring sites, the Beijing municipality (the areas within the blue line, and the surrounding regions. Also shown are locations of Tiananmen, and the $3^{rd}$, $4^{th}$, $5^{th}$ and $6^{th}$ ring roads.

**Fig. 2**. Concentrations of $NH_3$ (**A**) and $NO_2$ (**B**) during the monitoring periods at different observation scales: concentrations at 31 ($NH_3$) or 17 ($NO_2$) sites (a), averaged concentrations for the sites inside the $6^{th}$ ring (R) road (Rd), outside the 6th ring (R) road (Rd) and outside Beijing (b), averaged concentrations for the sites on the $3^{rd}$, $4^{th}$ and/or $5^{th}$ ring roads and non-road sites (NRS) (c) (one asterisk on bars denotes significant difference at $p<0.05$, two asterisks on bars denote significant difference at $p<0.01$).

**Fig. 3**. Comparison of $PM_{2.5}$, $PM_{10}$, $NO_2$, $SO_2$ and CO concentrations between non-Parade Blue periods (the pre- and post-Parade Blue periods) and Parade Blue period at Beijing, cities in emission control regions (excluding Beijing) and other cities in non-emission control regions (one asterisk on bars denotes significant difference at $p<0.05$, two asterisks on bars denote significant difference at $p<0.01$).

**Fig. 4**. Daily values of AQI and daily ratios of CO to $SO_2$ concentrations and of $PM_{2.5}$ to CO concentrations in Beijing during the pre-Parade Blue, Parade Blue and post-Parade Blue periods.

**Fig. 5**. Correlations between $NO_2$ and $NH_3$ concentrations measured on the $5^{th}$ ring road in Beijing during the pre-Parade Blue, Parade Blue, and post-Parade Blue periods.

**Fig. 6**. Comparison of wind speed (WS), relative humidity (RH) and temperature (*T*) between the Parade Blue period and non-Parade Blue periods (the pre-Parade Blue and post-Parade Blue periods) in Beijing, emission control regions (excluding Beijing) and other cities in non-emission control regions (two asterisk on bars denotes significant difference at $p<0.01$).

**Fig. 7**. Mean sea level pressure (unit: hPa) and mean wind field at 10 m height (unit:
m/s) during the pre-Parade Blue (a), Parade Blue (b) and post-Parade Blue (c)
periods in Beijing and North China. The color bar denotes air pressure (unit: hPa)
and arrows reflect wind vector (unit: m s$^{-1}$).
**Fig. 8**. The frequency distributions of wind directions and speeds (color demarcation)
(a), and daily precipitation amount (b) in Beijing city during the pre-Parade Blue,
Parade Blue, and post-Parade Blue periods.
**Fig. 9**. Dynamics of daily mean atmospheric mixing layer height (MLH) in Beijing
from 3 August to 30 September 2015 (a) and comparison of MLH means during the
pre-Parade Blue, Parade Blue and post-Parade Blue periods (b).
**Fig. 10**. 72-h backward trajectories for 100 m above ground level in Beijing city
during the pre-Parade Blue period (1 to 19 August 2015) (a), the Parade Blue period
(20 August to 3 September 2015) (b), and the post-Parade Blue period (4 to 30
September 2015) (c), and for sampling duration of NH$_3$ (8 to 19 September 2015) in
the post-Parade Blue period (d).
**Fig. 11** Effect of meteorological condition change (MCC, simulated by a
GEOS-Chem chemical transport model) and pollution control measures (PEM,
measured by monitoring stations) to relative concentrations of CO, NO$_2$, SO$_2$, NH$_3$
and PM$_{2.5}$ (a) and relative contribution of MCC and PEC to major pollutant
mitigation (b) in Beijing during the Parade Blue period.



**Figure 1**

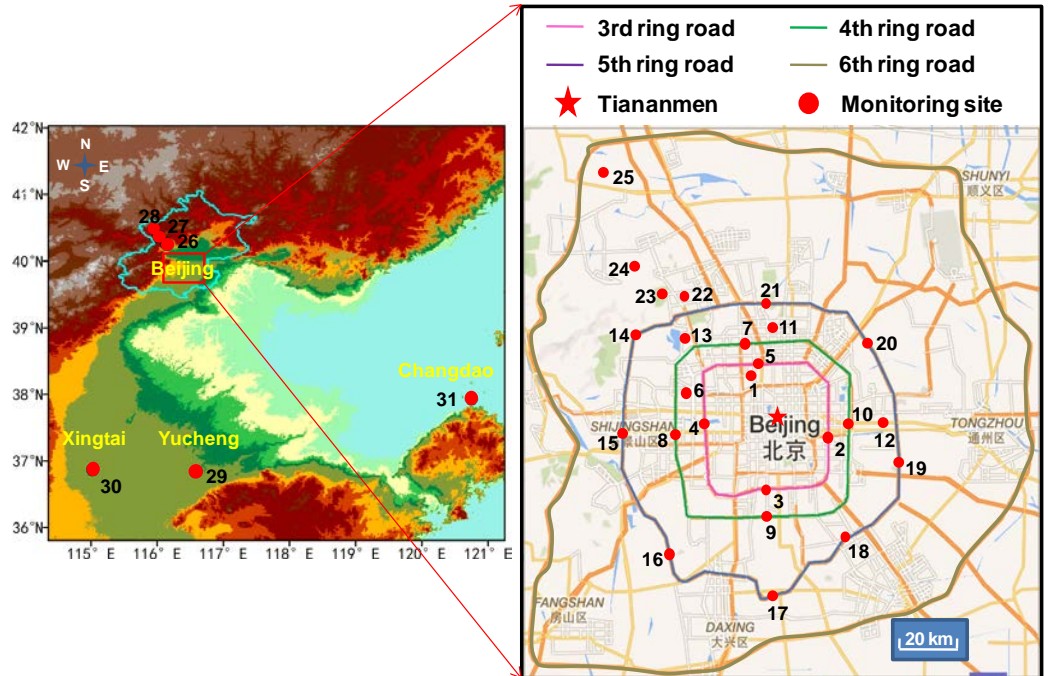


**Figure 2**

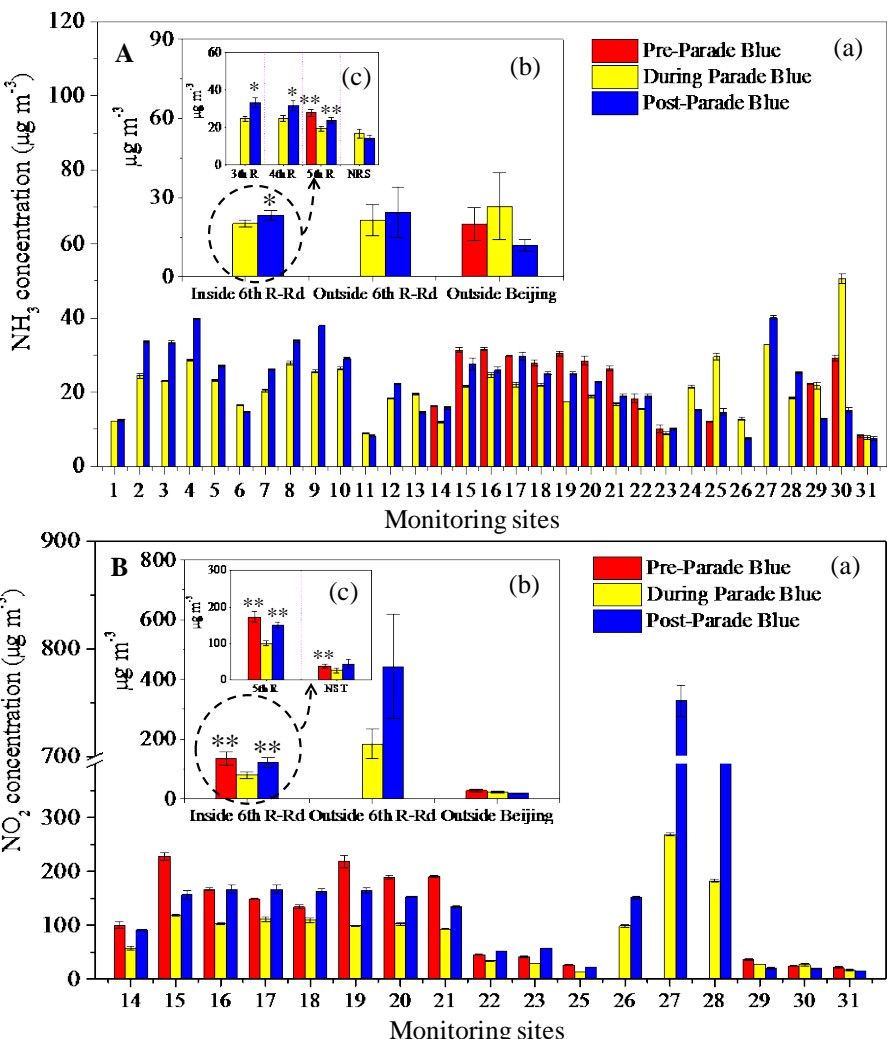

946    **Figure 3**

947


**Figure 4**


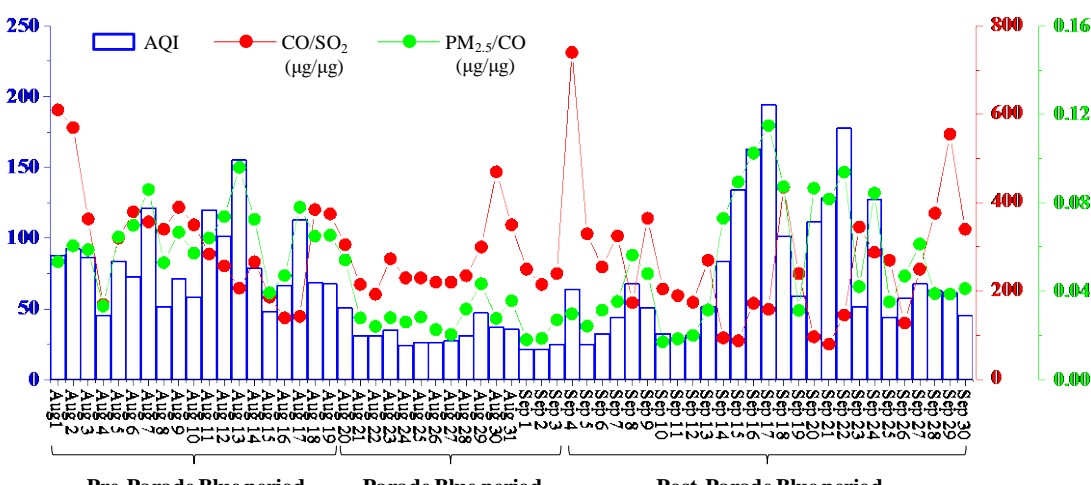













**Figure 5**

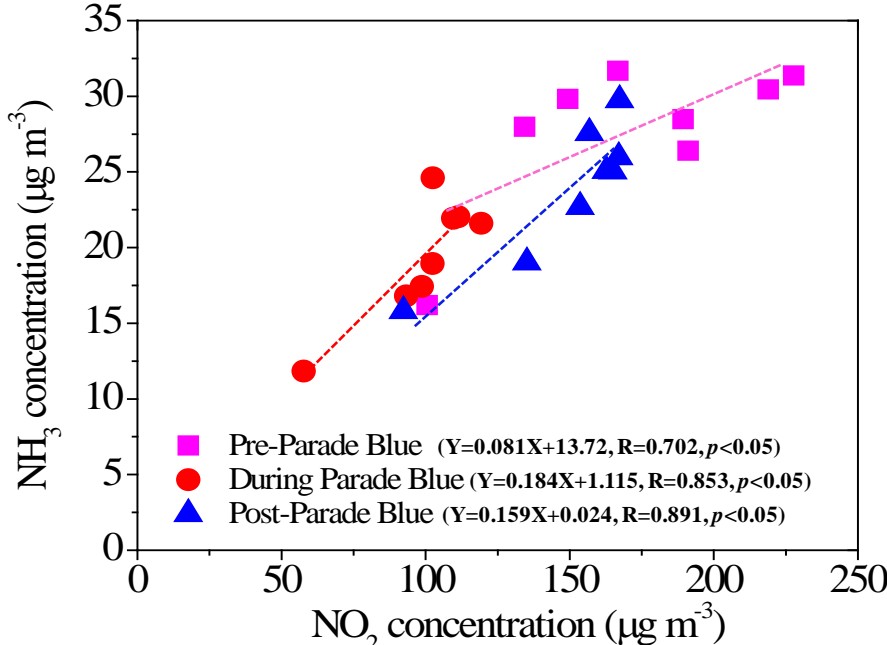

990     **Figure 6**

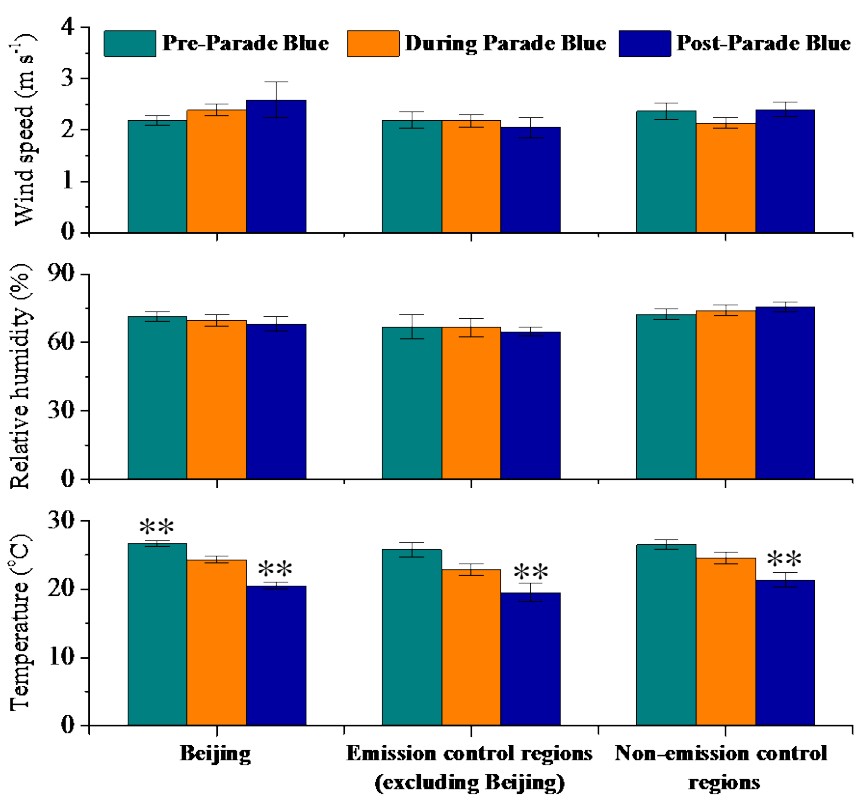

**Figure 7**

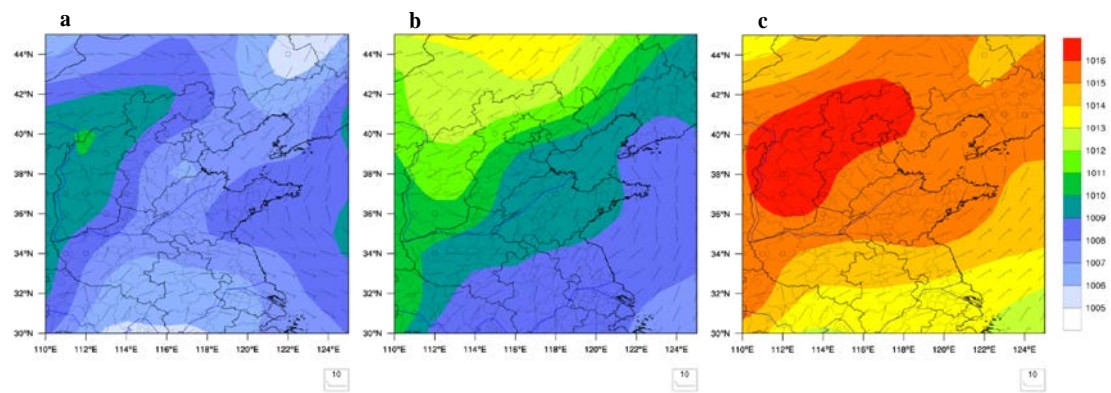






















**Figure 8**

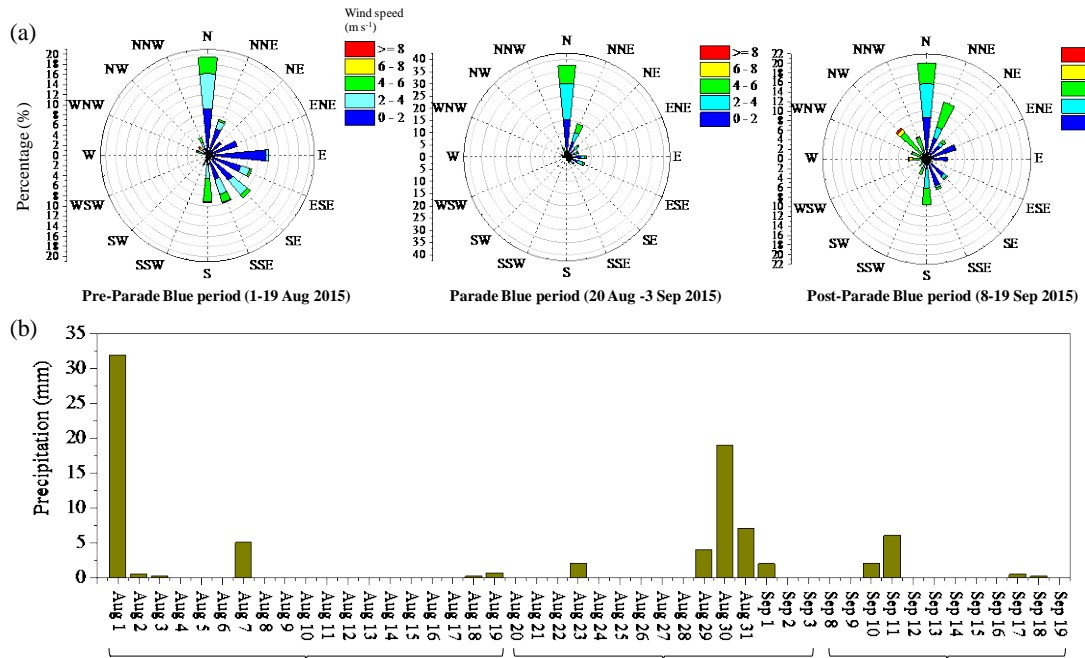



















**Figure 9**

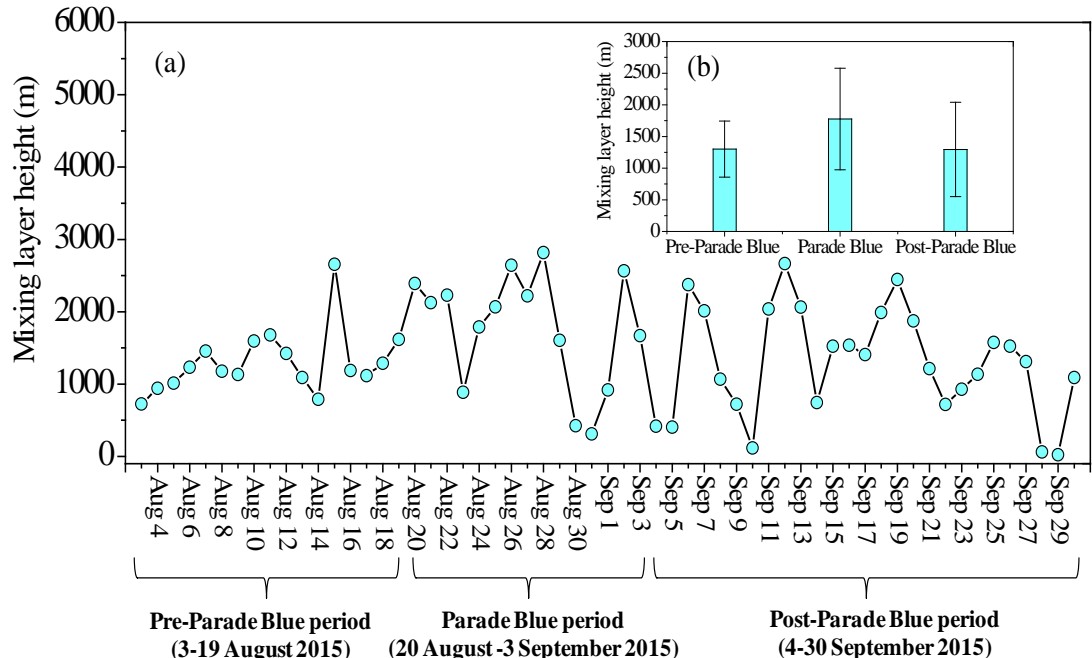



















**Figure 10**

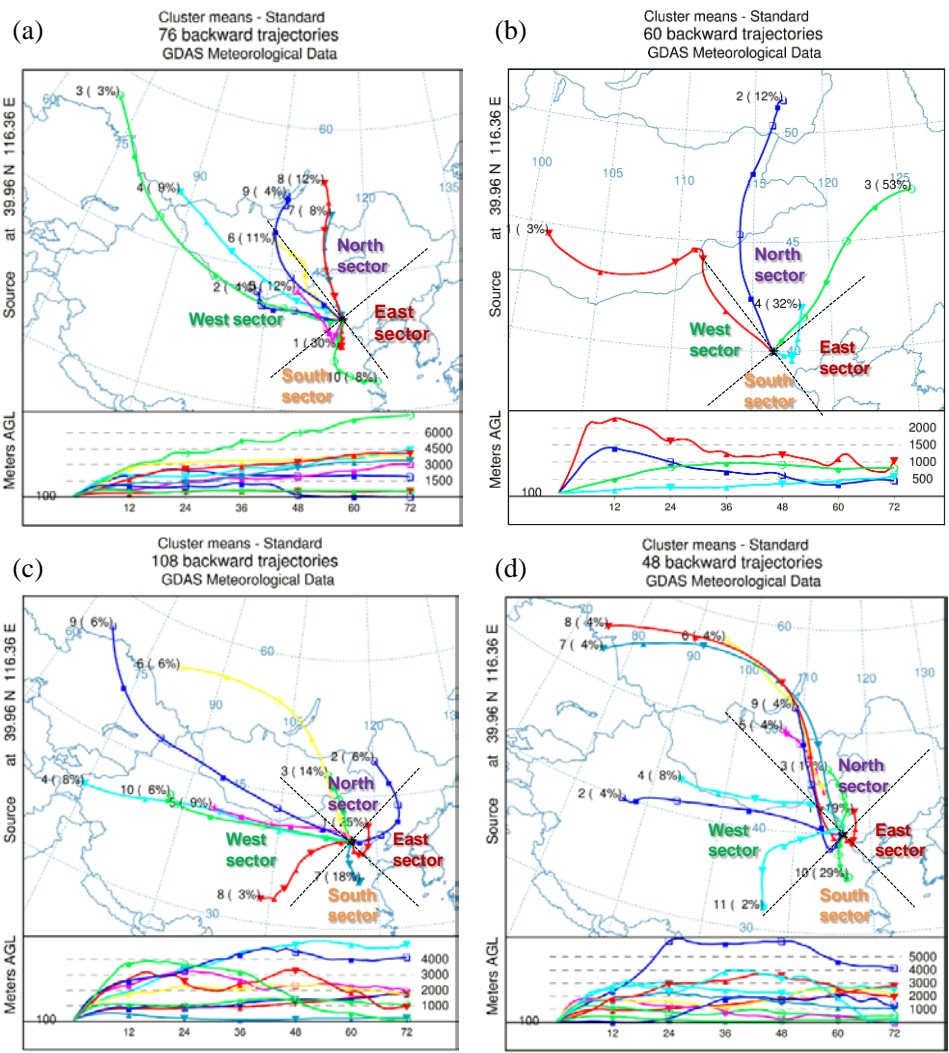













**Figure 11**

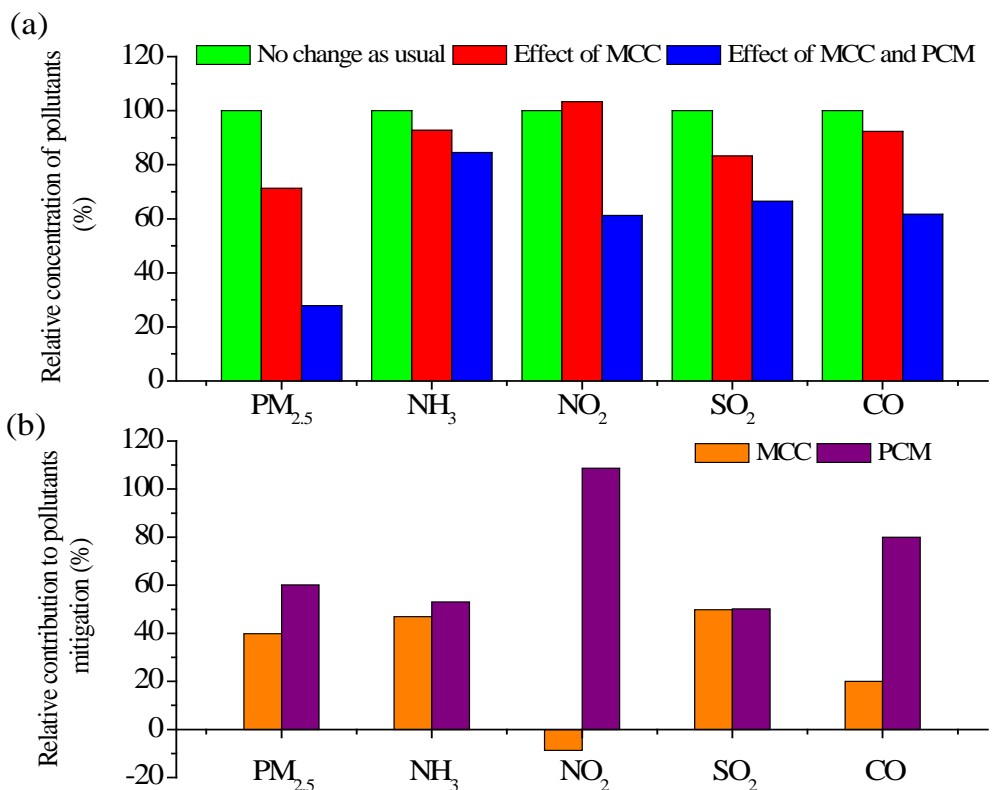
















**Table 1. Mean (SE) ambient concentrations of PM$_{2.5}$ and associated ionic components at the urban and rural sites.**

| | Urban site (Site 22) in Beijing | | | Rural site (Site 29) in Shandong | | | Rural site (Site 30) in Hebei | | |
|---|---|---|---|---|---|---|---|---|---|
| | Pre-PBP (n=11) | PBP[a] (n=15)[b] | Post-PBP (n=15) | Pre-PBP (n=6) | PBP (n=5) | Post-PBP (n=10) | Pre-PBP (n=6) | PBP (n=5) | Post-PBP (n=8) |
| PM$_{2.5}$ | 72.37 (7.36)** | 37.23 (5.37) | 58.49 (7.99) | 90.27 (8.53)* | 53.84 (11.37) | 55.30 (7.45) | 38.73 (5.17) | 29.44 (6.55) | 59.73 (16.35) |
| NO$_3^-$ | 2.07 (0.60) | 0.85 (0.15) | 6.27 (1.72)** | 4.21 (1.71) | 1.22 (0.22) | 5.56 (1.03)** | 0.58 (0.22) | 1.02 (0.05) | 3.46 (0.81)* |
| SO$_4^{2-}$ | 13.26 (2.85)** | 3.79 (0.69) | 10.92 (2.94) | 25.53 (3.36)* | 11.55 (3.20) | 14.80 (2.84) | 9.57 (1.07)* | 6.04 0.65 | 8.21 0.89 |
| NH$_4^+$ | 4.62 (0.94)** | 1.15 (0.26) | 4.07 (1.25) | 8.85 (0.91)* | 3.49 (1.01) | 4.32 (0.98) | 2.41 (0.30)** | 0.58 0.18 | 2.34 (0.40)** |
| Ca$^{2+}$ | 0.58 (0.04)** | 0.38 (0.06) | 0.51 (0.07) | 0.29 (0.06) | 0.29 (0.11) | 0.23 (0.05) | 0.19 (0.07) | 0.12 (0.02) | 0.09 (0.02) |
| K$^+$ | 0.30 (0.04)** | 0.15 (0.02) | 0.42 (0.08)** | 0.76 (0.07) | 0.50 (0.11) | 0.99 (0.18) | 0.20 (0.03) | 0.18 (0.02) | 0.24 (0.02) |
| F$^-$ | 0.17 (0.02)* | 0.10 (0.01) | 0.07 (0.02) | 0.04 (0.03) | 0.07 (0.03) | 0.10 (0.04) | 0.01 (0.00) | 0.00 (0.00) | 0.00 (0.00) |
| Cl$^-$ | 0.11 (0.01) | 0.11 (0.01) | 0.13 (0.03) | 0.14 (0.03) | 0.29 (0.14) | 0.19 (0.06) | 0.06 (0.03) | 0.01 (0.00) | 0.24 (0.09)* |
| Na$^+$ | 0.10 (0.02) | 0.09 (0.02) | 0.25 (0.05)** | 0.25 (0.05) | 0.45 (0.25) | 0.42 (0.04) | 0.35 (0.08) | 0.52 (0.06) | 0.26 (0.02)** |
| Mg$^{2+}$ | 0.08 (0.01)** | 0.05 (0.01) | 0.07 (0.01) | 0.05 (0.01) | 0.15 (0.12) | 0.07 (0.01) | 0.03 (0.00)** | 0.04 (0.00) | 0.04 (0.00) |
| SIA[c] | 19.95 (3.83)** | 5.78 (1.00) | 21.26 (5.83)* | 38.58 (3.75)** | 16.26 (4.19) | 24.68 (4.61) | 12.56 (1.43)* | 7.64 (0.81) | 14.00 (1.97)* |
| SIA/PM$_{2.5}$ (%) | 25.4 (3.2) | 20.0 (4.2) | 29.0 (4.8) | 42.9 (2.3) | 31.4 (3.7) | 45.6 (4.7) | 35.1 (5.2) | 30.4 (5.6) | 30.1 (4.4) |

[a] Parade Blue period. [b] Number of samples. [c] Secondary inorganic aerosol.

*Significant at the 0.05 probability level. **Significant at the 0.01 probability level.