# Peer review of "Air Quality Improvement in a Megacity: Implications from 2015 Beijing Parade"

_Atmospheric Chemistry and Physics, 2016_

## Referee Comment (RC1) · Anonymous Referee #3 · 30 Sep 2016

Air pollution in China, especially the North China, has drawn the world's attention in recent years. The Chinese government is capable of applying stringent and intensive emission control over a short period, during which an important event was held in Beijing or other major cities. This usually resulted in remarkably clean sky during the event. These manmade "control experiments" provide good opportunities to disentangle the complicated impacts of anthropogenic emissions and meteorological conditions on air quality. There have been quite a few published studies on the emission control periods of the 2008 Olympics, the 2014 APEC, and the 2015 Beijing Parade. This study distinguishes itself by analyzing wide spread monitoring sites both in Beijing and nation-wide (measurement network of $NO_2$ and $NH_3$ in Beijing and three NCP background sites is

certainly a plus) and using a GEOS-Chem model to quantify the relative contributions of emission reduction and favorable meteorological conditions to air quality improvement. Generally, this manuscript provides useful observation datasets and insightful analyses on the emission-control policy. I recommend this manuscript to be accepted by ACP if the following concerns can be addressed.

Major points:

1. At line 110, the authors stated that the previous studies did not systematically quantify the contribution of $NH_3$ from traffic sources to urban $PM_{2.5}$ and implied at they could address this question through their $NH_3$ observations. However, they only reported the concentration levels of $NH_3$ and stopped at the conclusion that on-road traffic is an important source of $NH_3$ in urban Beijing. It has long been demonstrated that on-road vehicles are major $NH_3$ sources for roadside sites. It is also obvious that reducing on-road vehicles during the emission control period will reduce $NH_3$ concentrations measured at these sites. It will require more analyses to extend from $NH_3$ concentrations measured at a few sites to $PM_{2.5}$ at the city scale. The authors brought up an important question about the linkage between $NH_3$ emission and $PM_{2.5}$ (which is one key reason why we care about $NH_3$), but did not really answer that. The authors should have the resources for further analyses, as the GEOS-Chem model can readily link the $NH_3$ emissions to $PM_{2.5}$.

2. Another innovative aspect of this study is characterizing the nonlinear response of pollutants to emission reduction (lines 111-112). The magnitude and spatial pattern of emission reductions are very important quantities, but they were assumed in a fairly arbitrary way (line 224). I think these assumptions need to be justified. In Fig. 13, one may argue that the response could be linear if the emission reductions are uncertain (say what assumed to be $5\%$ reduction is really $10\%$). Are these emission reductions also used in GEOS-Chem simulation?

3. The third innovation is quantifying the relative roles of emission reduction and favorable meteorology in air quality improvement (lines 112-113). The conclusion about this topic was drawn from the GEOS-Chem simulation, but I found that there is very limited information about this vital part of the analysis. What were the simulation time period and spatial coverage? What anthropogenic emission inventories were used? How were the emissions reduced in the model when simulating the Parade blue period? How did the simulation results compare with the rich observation datasets presented in Section 3?

Minor points:

1. Lines 91-92 and line 97: Is "North China" equivalent to the geographical extension of these six provinces or parts of them? The air mass and meteorology in parts of Inner Mongolia could be very different from the North China Plain.

2. Lines 139-141: Are these two sites (27 and 28) inside or outside the tunnel? $NO_2$ concentration at site 27 seems extremely high for ambient measurements (Fig. 2 B). Is that because it is in the tunnel?

3. Lines 201-202: Why was post-parade blue period not included in this dataset?

4. Lines 258-260: It will be less confusing if the acronyms (SWR, SOI, SOB) are spelled out.

5. Line 289: Please define WSI.

6. Line 441-442: Beijing actually has large agricultural sources, and its dominant $NH_3$ sources are still agriculture, at least according to the inventories. To argue that traffic is indeed an important $NH_3$ in Beijing, the authors need to provide more evidence on the roles of traffic emissions on $PM_{2.5}$ and/or on human/ecosystem exposures. See the first major comment point.

[Figure]

7. Line 451: I want to bring it to the authors' attention that Chang et al. (2016) reported mileage-based $NH_3$ emission factor of 28 mg/km in Shanghai, one order of magnitude smaller than the emission factor used here. Note that one of the coauthors of this work is also on the author list of Change et al. (2016).

8. Lines 452-456: These numbers do not mean that much, just multiplying literature emission factor and activity data. If the traffic $NH_3$ emission was reduced by a half, can it explain the observed reduction in $NH_3$ concentrations? Did $NH_3$ emission reduction play any role in the $PM_{2.5}$ reduction?

9. Line 517: Why was Fig. 4 mentioned after Figs. 5–11?

10. Figure 2: Please define the meaning of "*" and "**" in the caption.

11. Figure 8: This figure is hard to read and seems redundant with Fig. 9 for the wind and pressure.

12. Table S1 caption: Information on the "thirty-one" monitoring sites?

Reference:

Chang, Y., Zou, Z., Deng, C., Huang, K., Collett, J. L., Lin, J. and Zhuang, G.: The importance of vehicle emissions as a source of atmospheric ammonia in the megacity of Shanghai, Atmos. Chem. Phys., 16(5), 3577–3594, doi:10.5194/acp-16-3577-2016, 2016.

---

## Referee Comment (RC2) · Anonymous Referee #1 · 3 Nov 2016

Recommend publication as it is.

---

## Author Comment (AC1) · 11 Dec 2016

**Anonymous Referee #1**

**General comments**

Recommend publication as it is.

**Response:** We thank the reviewer's recommendation for publication.

---

## Author Comment (AC2) · 11 Dec 2016

**Anonymous Referee [#]3**
**General comments**

Air pollution in China, especially the North China, has drawn the world's attention in recent years. The Chinese government is capable of applying stringent and intensive emission control over a short period, during which an important event was held in Beijing or other major cities. This usually resulted in remarkably clean sky during the event. These manmade "control experiments" provide good opportunities to disentangle the complicated impacts of anthropogenic emissions and meteorological conditions on air quality. There have been quite a few published studies on the emission control periods of the 2008 Olympics, the 2014 APEC, and the 2015 Beijing Parade. This study distinguishes itself by analyzing wide spread monitoring sites both in Beijing and nation-wide(measurement network of $NO_2$ and $NH_3$ in Beijing and three NCP background sites is certainly a plus) and using a GEOS-Chem model to quantify the relative contributions of emission reduction and favorable meteorological conditions to air quality improvement. Generally, this manuscript provides useful observation datasets and insightful analyses on the emission-control policy. I recommend this manuscript to be accepted by ACP if the following concerns can be addressed.

**Response:** We thank the reviewer for the encouraging words and insightful comments. We believe that addressing the issues pointed out by the reviewer will considerably improve the manuscript. Please see our itemized responses below.

**Major points:**

1. At line 110, the authors stated that the previous studies did not systematically quantify the contribution of $NH_3$ from traffic sources to urban $PM_{2.5}$ and implied at they could address this question through their $NH_3$ observations. However, they only reported the concentration levels of $NH_3$ and stopped at the conclusion that on-road traffic is an important source of $NH_3$ in urban Beijing. It has long been demonstrated that on-road vehicles are major $NH_3$ sources for roadside sites. It is also obvious that reducing on-road vehicles during the emission control period will reduce $NH_3$ concentrations measured at these sites. It will require more analyses to extend from

NH$_3$ concentrations measured at a few sites to PM$_{2.5}$ at the city scale. The authors brought up an important question about the linkage between NH$_3$ emission and PM$_{2.5}$ (which is one key reason why we care about NH$_3$), but did not really answer that. The authors should have the resources for further analyses, as the GEOS-Chem model can readily link the NH$_3$ emissions to PM$_{2.5}$.

**Response:** Thank you for your comment. A main objective of this study is to quantify the changes in air quality during the 2015 Beijing Parade Period as evidenced by our field measurements. The GEOS-Chem model simulation (with fixed anthropogenic emissions during the period) was used to provide an estimate of the relative role of meteorology. We do not include a detailed model analysis, since that should be done as a separate study that requires extensive work on evaluation of the model emissions and chemical mechanism.

To address the comment, we have made the following revisions: 1) we changed in the sentence "the contribution of ammonia (NH$_3$) from traffic sources to urban PM$_{2.5}$ pollution" to "the contribution of ammonia (NH$_3$) sources to urban PM$_{2.5}$ pollution"; 2) we examined the impacts of NH$_3$ emissions on urban PM$_{2.5}$ concentrations using GEOS-Chem model sensitivity simulations with perturbed anthropogenic NH$_3$ emissions in the Beijing-Tianjin-Hebei region. Linking NH$_3$ from traffic sources alone to PM$_{2.5}$ concentrations is a challenging task in the model, as a proper estimate of traffic NH$_3$ emission inventory is lacking. We have added and presented the model results in Sect. 4.3, as summarized below in addressing the comment #3.

2. Another innovative aspect of this study is characterizing the nonlinear response of pollutants to emission reduction (lines 111-112). The magnitude and spatial pattern of emission reductions are very important quantities, but they were assumed in a fairly arbitrary way (line 224). I think these assumptions need to be justified. In Fig. 13, one may argue that the response could be linear if the emission reductions are uncertain (say what assumed to be 5% reduction is really 10%). Are these emission reductions also used in GEOS-Chem simulation?

**Response:** In the revised manuscript, we now remove this part (discussion of

non-linear response to emission reduction; including Figure 13 and several sentences in Sect. 4.3). This revision does not change the structure of the paper. We agree that characterizing the nonlinear response of air pollutants to emission reduction based on field measurements may have considerable uncertainties, and we think a detailed model investigation of this feature is beyond the scope of this study (but could be done in a future study).

3. The third innovation is quantifying the relative roles of emission reduction and favorable meteorology in air quality improvement (lines 112-113). The conclusion about this topic was drawn from the GEOS-Chem simulation, but I found that there is very limited information about this vital part of the analysis. What were the simulation time period and spatial coverage? What anthropogenic emission inventories were used? How were the emissions reduced in the model when simulating the Parade blue period? How did the simulation results compare with the rich observation datasets presented in Section 3?

**Response:** We examined the relative roles of emission reduction and meteorology in Section 4.3 by first presenting a comprehensive analysis of the meteorological conditions (temperature, RH, wind patterns, and precipitations) before/during/after the Parade Blue period, and then providing a quantitative estimate using the GEOS-Chem model simulation. In the revised manuscript we have added the following text to more fully describe and discuss the GEOS-Chem simulation:

"To further diagnose the impacts of meteorology on the surface air quality, we conducted a simulation using the nested GEOS-Chem atmospheric chemistry model driven by the GEOS-FP assimilated meteorological fields at 1/4°×5/16° horizontal resolution covering East Asia (70°E-140°E, 15°N-55°N) (Zhang et al., 2015; 2016). Details of the model emissions and mechanisms have been described in Zhang et al. (2016), focusing on $PM_{2.5}$ concentrations in Beijing during the Asia-Pacific Economic Cooperation Summit (APEC; November 5-11) period. We used anthropogenic emissions from the Multi-Resolution Emission Inventory of China for the year 2010 (MEIC, 2015), except for $NH_3$ emissions that are taken from the Regional Emission in

Asia (REAS-v2) inventory (Kurokawa et al., 2013) with an improved seasonality derived by Zhao et al. (2015)."

"We conducted a standard simulation with fixed anthropogenic emissions for the period of 1 August – 12 September 2015. By fixing anthropogenic emissions in the simulation, the model provides a quantitative estimate of the meteorological impacts alone before and during the Parade Blue period. For the pre-Parade period (1-19 August), the model simulated mean $PM_{2.5}$ concentration is 62 µg m$^{-3}$ in Beijing, comparable to the measured values (59 µg m$^{-3}$), but simulated $NH_3$ concentrations are too low (3 µg m$^{-3}$ vs. measured values of 8.2-31.7 µg m$^{-3}$), probably due to missing urban $NH_3$ sources and the coarse model resolution (1/4°×5/16°). Here we focus on the model simulated relative changes in pollutant concentrations before and during the Parade Blue period."

"We also conducted two sensitivity simulations ((1) with anthropogenic emissions of $NH_3$ reduced by 40% over Beijing and by 30% over Hebei and Tianjin; and (2) with all anthropogenic emissions including $NH_3$, $SO_2$, $NO_x$, CO, and primary aerosol reduced by 40% over Beijing and by 30% over Hebei and Tianjin) for the Parade Blue period (20 August-3 September 2015) to examine the responses of $PM_{2.5}$ concentrations to emission reductions. We find that the $NH_3$ emission reduction (by 40% over Beijing and by 30% over Hebei and Tianjin) could decrease the mean $PM_{2.5}$ concentration in Beijing by 12% for the period, compared with 31% simulated $PM_{2.5}$ reduction if all anthropogenic emissions were reduced by the same amount. This supports our findings on the effectiveness of emission controls during the Parade Blue period as indicated in the measurements, and the high sensitivity of $PM_{2.5}$ concentration in Beijing to $NH_3$ sources."

**Minor points:**

1. Lines 91-92 and line 97: Is "North China" equivalent to the geographical extension of these six provinces or parts of them? The air mass and meteorology in parts of Inner Mongolia could be very different from the North China Plain.

**Response:** North China covers parts of these six provinces. According to the typical

division of meteorological geographical areas in China, North China includes Beijing City, Tianjing City, the middle area of Inner Mongolia (i.e., Hohhot, Baotou, Ordos and Ulanchap Cities), Hebei, Shanxi, and Shandong provinces. To avoid misunderstanding, we deleted "North China" in line 90, and instead used "surrounding regions" and/or "emission control regions (excluding Beijing)" for related expressions throughout the entire revised manuscript.

2. Lines 139-141: Are these two sites (27 and 28) inside or outside the tunnel? $NO_2$ concentration at site 27 seems extremely high for ambient measurements (Fig. 2B). Is that because it is in the tunnel?

**Response:** Yes, sites 27 and 28 were located, respectively, inside and outside the tunnel. We have revised the sentence (lines 139-141) and now it reads: "Sites 27 and 28 are located, respectively, inside (100 m from the exit) and outside (30 m from the entrance) the Badaling Highway Tunnel (1091.2 m long), which has two traffic tunnels with one lane in each."

The extremely high concentrations of $NO_2$ measured at site 27 were most likely due to enhanced $NO_2$ accumulation in the tunnel resulting from limited dispersion of pollutants. A similar phenomenon was also observed for $NH_3$ concentrations as reported by Chang et al. (2016).

3. Lines 201-202: Why was post-parade blue period not included in this dataset?

**Response:** We chose the pre-Parade Blue and Parade Blue periods for our comparison partly because the changes in concentrations of the pollutants between these two periods can achieve the aims of the study. We now add the 24-h (daily) average concentrations of $PM_{2.5}$, $PM_{10}$, $NO_2$, $SO_2$ and CO during the post-Parade Blue period in the revised manuscript, as shown in improved Figure 3 and also summarized in Tables S2-S6 in the Supplement. Accordingly, additional information during the post-Parade Blue period was also added in order to further improve the manuscript, including Air Quality Index (AQI), ratios of $CO/SO_2$, and $PM_{2.5}/CO$ (Figure 4), meteorological parameters (wind speed, wind direction and relative humidity) (Figure 6), sulfur oxidation ratio (SOR) and nitrogen oxidation ratio (NOR) (Figure S4 of the

Supplement), correlations between $PM_{2.5}$ and gaseous pollutants (e.g. CO, $NO_2$, $SO_2$) (Figure S2 of the Supplement), and correlations between $NH_4^+$, $SO_4^{2-}$, and their sum (Figure S5 of the Supplement).

[Figure]

Figure 3. Comparison of $PM_{2.5}$, $PM_{10}$, $NO_2$, $SO_2$ and CO concentrations between non-Parade Blue periods (the pre- and post-Parade Blue periods) and Parade Blue period at Beijing, cities in emission control regions (excluding Beijing) and other cities in non-emission control regions (one asterisk on bars denotes significant difference at $p<0.05$, two asterisks on bars denote significant difference at $p<0.01$).

4. Lines 258-260: It will be less confusing if the acronyms (SWR, SOI, SOB) are spelled out.

**Response:** We accept this suggestion and have spelled out the acronyms (SWR, SOI, SOB) in the revised paper.

5. Line 289: Please define WSI.

**Response: WSI represents water-soluble ions.** We have defined WSI in Line 118 where it first appears.

6. Line 441-442: Beijing actually has large agricultural sources, and its dominant $NH_3$ sources are still agriculture, at least according to the inventories. To argue that traffic is indeed an important $NH_3$ in Beijing, the authors need to provide more evidence on the roles of traffic emissions on $PM_{2.5}$ and/or on human/ecosystem exposures. See the first major comment point.

**Response:** Thank you for this suggestion, but we are skeptical about the reviewer's view that dominant $NH_3$ sources in Beijing are still agriculture. Recently, some studies have discussed the origin of atmospheric $NH_3$ in Beijing based on the $\delta^{15}N$ technique (Chang et al., 2016; Pan et al., 2016). Chang et al. (2016) identified that non-agricultural sources, merged with waste and traffic $NH_3$ emissions, collectively accounted for approximately 50% of ambient $NH_3$ in urban Beijing before and after APEC summit, of which more than 20% was sourced from traffic emissions. Pan et al. (2016) claimed that fossil fuel-based $NH_3$ emissions (including traffic, coal combustion and power plants) have overtaken agricultural emissions as the dominant source of atmospheric $NH_3$ during the hazy days in urban Beijing. Such results from those studies cannot be explained by previous emission inventories (e.g., Fu et al., 2013; Huang et al., 2011, 2012; Kuang et al., 2016; M. Li et al., 2015; Zhang et al., 2009, 2010). Thus the attribution of $NH_3$ sources in Beijing is still an open topic. In contrast to the studies of Chang et al. (2016) and Pan et al. (2016), the current study directly measured $NH_3$ and $NO_2$ concentrations at road sites. Our results show positive and significant relationships between $NH_3$ and $NO_2$ during different monitoring periods, and average $NH_3$ concentrations for all three ring roads decreased significantly during the Parade Blue period when compared with the pre- and post-Parade Blue period. In addition, during the post-Parade Blue period the measured $NH_3$ concentrations on the three ring roads (28.3 ±6.4 μg m$^{-3}$) were twice those at the rural sites 29 and 30 (14.0 ± 1.6 μg m$^{-3}$) affected by intense agricultural $NH_3$ emissions. We think that the preceding discussion results (Section 4.2) are sufficient to confirm that traffic is indeed an important $NH_3$ source in Beijing.

The above discussions have been briefly summarized in the revised paper. In addition,

we have conducted model sensitivity simulations to examine the impacts of anthropogenic $NH_3$ emissions on $PM_{2.5}$ concentration at Beijing, and presented the model results in the text, as described in the response to the first comment above.

7. Line 451: I want to bring it to the authors' attention that Chang et al. (2016)reported mileage-based $NH_3$ emission factor of 28 mg/km in Shanghai, one order of magnitude smaller than the emission factor used here. Note that one of the coauthors of this work is also on the author list of Chang et al. (2016).

**Response:** Thank you for this comment. The authors also found that the emission factor for vehicle-emitted $NH_3$ by Chang et al. (2016) was an order of magnitude smaller than that by Liu et al. (2014) which was used in the present study, but was similar to that estimated for the Gurbrist tunnel in Switzerland ($31\pm4$ mg $km^{-1}$) (Emmenegger et al., 2004) and the Caldecott tunnel in California ($49\pm3$ mg $km^{-1}$) (Kean et al., 2000). Different results obtained from those studies can be partially explained by differences in vehicle and fuel types, emission control technology, driving patterns and experimental methods (Keen et al., 2000, 2009; Baum et al., 2001; Heeb et al., 2006, 2008; Livingston et al., 2009). Therefore, we cannot judge the accuracy of existing emission factors of vehicle-emitted $NH_3$. In the revised paper, we assumed that $NH_3$ emission factors by Chang et al. (2016) and Liu et al. (2014) can both be representative for Beijing, and then used them to estimate a range for the amount of traffic-related $NH_3$ emissions during the Parade Blue period. For accurately determining $NH_3$ emissions, however, further study on $NH_3$ emission factors for vehicles at the Badaling Tunnel is warranted.

8. Lines 452-456: These numbers do not mean that much, just multiplying literature emission factor and activity data. If the traffic $NH_3$ emission was reduced by a half, can it explain the observed reduction in $NH_3$ concentrations? Did $NH_3$ emission reduction play any role in the $PM_{2.5}$ reduction?

**Response:** We agree with the reviewer that the estimate of $NH_3$ emission based on a literature emission factor and activity data does not mean that much. However, we also think that it would be meaningful to give a quantitative formation for the reader,

albeit with some uncertainty. As mentioned before (see response to Comment #7), our future work will address this issue based on realistic estimate of $NH_3$ emission factors for vehicles at the Badaling Tunnel and other tunnels.

According to $NH_3$ measurements in the present study, the mean $NH_3$ concentration at all sites within the $6^{th}$ ring road in Beijing was significantly decreased (by 13%) during the Parade Blue period (when traffic load decreased by half) compared with the mean during the post-Parade Blue period without implementation of the odd-and-even car ban policy; further, on all three ring roads significant reductions (23 to 35%) of the mean during the Parade Blue period were also observed. Therefore, reduction of traffic $NH_3$ emission by half can partially explain the observed reduction in $NH_3$ concentrations, as emitted $NH_3$ in the atmosphere can be affected by many factors, such as acid gaseous ($SO_2$ and $NO_2$), meteorological conditions (e.g. wind speed, wind direction, precipitation) as well as regional transport (Meng et al., 2011; Behera et al., 2013).

Also for addressing the first three comments above, we have now discussed in the revised manuscript the impacts of $NH_3$ emissions on $PM_{2.5}$ concentrations using the GEOS-Chem sensitivity simulations.

9. Line 517: Why was Fig. 4 mentioned after Figs. 5–11?

**Response:** We incorrectly mentioned the Fig. 4, and we have re-ordered the Figures in the revised paper.

10. Figure 2: Please define the meaning of "*" and "**" in the caption.

**Response:** Suggestion has been implemented.

11. Figure 8: This figure is hard to read and seems redundant with Fig. 9 for the wind and pressure.

**Response:** We agree with this comment. In contrast to Figure 9, Figure 8 shows period-to-period changes in wind field, sea surface pressure and precipitation over the whole China, which is helpful to interpret corresponding changes in pollutant concentrations in non-emission control regions and crucial for the reader. Therefore, we decide to keep Figure 8 but move it to the Supplement (see Figure S6) to avoid

repetitive description of wind and pressure over Beijing and emission controlled region in China.

12. Table S1 caption: Information on the "thirty-one" monitoring sites?

**Response:** Yes, we are sorry for this misspelling, and we have corrected in the revised version.

**Reference:**

Baum, M. M., Kiyomiya, E. S., Kumar, S., Lappas, A. M., Kapinus, V. A., and Lord, H. C.: Multicomponent remote sensing of vehicle exhaust by dispersive absorption spectroscopy. 2. Direct on-road ammonia measurements, Environ. Sci. Technol., 35, 3735-3741, doi:10.1021/Es002046y, 2001.

Behera, S. N., Sharma, M., Aneja, V. P., and Balasubramanian, R.: Ammonia in the atmosphere: a review on emission sources, atmospheric chemistry and deposition on terrestrial bodies, Environ. Sci. Pollut. R., 20, 8092–8131, doi:10.1007/s11356-013-2051-9, 2013.

Chang, Y., Zou, Z., Deng, C., Huang, K., Collett, J. L., Lin, J., and Zhuang, G.: The importance of vehicle emissions as a source of atmospheric ammonia in the megacity of Shanghai, Atmos. Chem. Phys., 16, 3577-3594, doi:10.5194/acp-16-3577-2016, 2016.

Emmenegger, L., Mohn, J., Sigrist, M., Marinov, D., Steinemann, U., Zumsteg, F., and Meier, M., Measurement of ammonia emissions using various techniques in a comparative tunnel study, Int. J. Environ. Pollut., 22, 326-341, doi:10.1504/IJEP.2004.005547, 2004.

Fu, X., Wang, S., Zhao, B., Xing, J., Cheng, Z., Liu, H., and Hao, J.: Emission inventory of primary pollutants and chemical speciation in 2010 for the Yangtze River Delta region, China, Atmos. Environ., 70, 39-50, doi:10.1016/j.atmosenv.2012.12.034, 2013.

Heeb, N. V., Saxer, C. J., Forss, A. M., and Brühlmann, S.: Correlation of hydrogen, ammonia and nitrogen monoxide (nitric oxide) emissions of gasoline-fueled euro-3 passenger cars at transient driving, Atmos. Environ., 40, 3750-3763, doi:10.1016/j.atmosenv.2006.03.002, 2006.

Heeb, N. V., Saxer, C. J., Forss, A. M., and Brühlmann, S.: Trends of NO-, $NO_2$-, and $NH_3$-emissions from gasoline-fueled euro-3-to euro-4-passenger cars, Atmos. Environ., 42, 2543-2554, doi:10.1016/j.atmosenv.2007.12.008, 2008.

Huang, C., Chen, C. H., Li, L., Cheng, Z., Wang, H. L., Huang, H. Y., Streets, D. G., Wang, Y. J., Zhang, G. F., and Chen, Y. R.: Emission inventory of anthropogenic air pollutants and VOC species in the Yangtze River Delta region, China, Atmos. Chem. Phys., 11, 4105-4120, doi:10.5194/acp-11-4105-2011, 2011.

Huang, X., Song, Y., Li, M., Li, J., Huo, Q., Cai, X., Zhu, T., Hu, M., and Zhang, H.: A high-resolution ammonia emission inventory in China, Global Biogeochem. Cy., 26, GB1030, doi:10.1029/2011GB004161, 2012.

Kean, A. J., Harley, R. A., Littlejohn, D., and Kendall, G. R.: On-road measurement of ammonia and other motor vehicle exhaust emissions, Environ. Sci. Technol., 2000, 34, 3535-3539, doi:10.1021/es991451q.

Kean, A. J., Littlejohn, D., Ban-Weiss, G. A., Harley, R. A., Kirchstetter, T. W., and Lunden, M. M.: Trends in on-road vehicle emissions of ammonia, Atmos. Environ., 43, 1565-1570, doi: 10.1016/j.atmosenv.2008.09.085.

Kurokawa, J., Ohara, T., Morikawa, T., Hanayama, S., JanssensMaenhout, G., Fukui, T., Kawashima, K., and Akimoto, H.: Emissions of air pollutants and greenhouse gases over Asian regions during 2000-2008: Regional Emission inventory in Asia (REAS) version 2, Atmos. Chem. Phys., 13, 11019-11058, doi:10.5194/acp-13-11019-2013, 2013.

Livingston, C., Rieger, P., and Winer, A.: Ammonia emissions from a representative in-use fleet of light and medium-duty vehicles in the California south coast air basin, Atmos. Environ., 43, 3326-3333, doi:org/10.1016/j.atmosenv.2009.04.009, 2009.

Li, M., Zhang, Q., Kurokawa, J., Woo, J. H., He, K. B., Lu, Z., Ohara, T., Song, Y., Streets, D. G., Carmichael, G. R., Cheng, Y. F., Hong, C. P., Huo, H., Jiang, X. J., Kang, S. C., Liu, F., Su, H., and Zheng, B.: MIX: a mosaic Asian anthropogenic emission inventory for the MICS-Asia and the HTAP projects, Atmos. Chem. Phys. Discuss., 15, 34813–34869, doi:10.5194/acpd-15-34813-2015, 2015.

Liu, T., Wang, X., Wang, B., Ding, X., Deng, W., Lü, S., and Zhang, Y.: Emission factor of ammonia (NH3) from on-road vehicles in China: Tunnel tests in urban Guangzhou, Environ. Res. Lett., 9, 064027, doi:10.1088/1748-9326/9/6/064027, 2014.

Meng, Z. Y., Lin, W. L., Jiang, X. M., Yan, P., Wang, Y., Zhang, Y. M., Jia, X. F., and Yu, X. L.: Characteristics of atmospheric ammonia over Beijing, China, Atmos. Chem. Phys., 11, 6139-6151, doi:10.5194/acp-11-6139-2011, 2011.

Pan, Y. P., Tian, S. L., Liu, D. W., Fang, Y. T., Zhu, X. Y., Zhang, Q., Zheng, B., Michalski, G., and Wang, Y. S.: Fossil Fuel Combustion-Related Emissions Dominate Atmospheric Ammonia Sources during Severe Haze Episodes: Evidence from $^{15}$N-Stable Isotope in Size-Resolved Aerosol Ammonium, Environ. Sci. Technol., 50, 8049-8056, doi: 10.1021/acs.est.6b00634, 2016.

Zhang, L., Liu, L. C., Zhao, Y. H., Gong, S. L., Zhang, X. Y., Henze, D. K., Capps, S. L., Fu, T. M., Zhang, Q., and Wang, Y. X.: Source attribution of particulate matter pollution over North China with the adjoint method, Environ. Res. Lett., 10, 084011, doi:10.1088/1748-9326/10/8/084011, 2015.

Zhang, L., Shao, J. Y., Lu, X., Zhao, Y. H., Hu, Y. Y., Henze, D. K., Liao, H., Gong, S. L., and Zhang, Q.: Sources and processes affecting fine particulate matter pollution over North China: an adjoint analysis of the Beijing APEC period, Environ. Sci. Technol., 50, 8731-8740, doi: 10.1021/acs.est.6b03010, 2016.

Zhang, Q., Streets, D. G., Carmichael, G. R., He, K. B., Huo, H., Kannari, A., Klimont, Z., Park, I. S., Reddy, S., Fu, J. S., Chen, D., Duan, L., Lei, Y., Wang, L. T., and Yao, Z. L.: Asian emissions in 2006 for the NASA INTEX-B mission, Atmos. Chem. Phys., 9, 5131–5153, doi:10.5194/acp-9-5131-2009, 2009.

Zhang, Y., Dore, A. J., Ma, L., Liu, X. J., Ma, W. Q., Cape, J. N., and Zhang, F. S.: Agricultural ammonia emissions inventory and spatial distribution in the North China Plain, Environ. Pollut., 158, 490-501, doi:10.1016/j.envpol.2009.08.033, 2010.

Zhao, Y., Zhang, L., Pan, Y., Wang, Y., Paulot, F., and Henze, D. K.: Atmospheric nitrogen deposition to the northwestern Pacific: seasonal variation and source

attribution, Atmos. Chem. Phys., 15, 10905-10924, doi:10.5194/acp-15-10905-2015, 2015.

---

## Author Comment (AC3) · 11 Dec 2016

**Anonymous Referee #1**

**General comments**

Recommend publication as it is.

**Response:** We thank the reviewer's recommendation for publication.